# Kicking sleepers out of bed: Macrophages promote reactivation of dormant *Cryptococcus neoformans* by extracellular vesicle release and non-lytic exocytosis

**Raffael Júnio Araújo de Castro**[1,2], **Clara Luna Marina**[2], **Aude Sturny-Leclère**[1], **Christian Hoffmann**[3], **Pedro Henrique Bürgel**[2], **Sarah Sze Wah Wong**[4], **Vishukumar Aimanianda**[4], **Hugo Varet**[5], **Ruchi Agrawal**[1], **Anamélia Lorenzetti Bocca**[2], **Alexandre Alanio**[1,6]*

**1** Translational Mycology Research Group, National Reference Center for Invasive Mycoses and Antifungals, Mycology Department, Institut Pasteur, Université Paris Cité, Paris, France, **2** Laboratory of Applied Immunology, Department of Cell Biology, Institute of Biological Sciences, University of Brasilia, Brasília, Distrito Federal, Brazil, **3** Food Research Center, Department of Food Sciences and Experimental Nutrition, Faculty of Pharmaceutical Sciences, University of São Paulo, São Paulo, São Paulo, Brazil, **4** Immunobiology of *Aspergillus*, Institut Pasteur, Université Paris Cité, Paris, France, **5** Plate-forme Technologique Biomics, Institut Pasteur, Université Paris Cité, Paris, France, **6** Laboratoire de parasitologie-mycologie, AP-HP, Hôpital Saint-Louis, Paris, France

* alexandre.alanio@pasteur.fr

**Data Availability Statement:** All sequencing data are available through the NCBI GEO database under

## Abstract

Macrophages play a key role in disseminated cryptococcosis, a deadly fungal disease caused by *Cryptococcus neoformans*. This opportunistic infection can arise following the reactivation of a poorly characterized latent infection attributed to dormant *C. neoformans*. Here, we investigated the mechanisms underlying reactivation of dormant *C. neoformans* using an *in vitro* co-culture model of viable but non-culturable (VBNC; equivalent of dormant) yeast cells with bone marrow-derived murine macrophages (BMDMs). Comparative transcriptome analysis of BMDMs incubated with log, stationary phase or VBNC cells of *C. neoformans* showed that VBNC cells elicited a reduced transcriptional modification of the macrophage but retaining the ability to regulate genes important for immune response, such as NLRP3 inflammasome-related genes. We further confirmed the maintenance of the low immunostimulatory capacity of VBNC cells using multiplex cytokine profiling, and analysis of cell wall composition and dectin-1 ligands exposure. In addition, we evaluated the effects of classic (M1) or alternative (M2) macrophage polarization on VBNC cells. We observed that intracellular residence sustained dormancy, regardless of the polarization state of macrophages and despite indirect detection of pantothenic acid (or its derivatives), a known reactivator for VBNC cells, in the *C. neoformans*-containing phagolysosome. Notably, M0 and M2, but not M1 macrophages, induced extracellular reactivation of VBNC cells by the secretion of extracellular vesicles and non-lytic exocytosis. Our results indicate that VBNC cells retain the low immunostimulatory profile required for persistence of *C. neoformans* in the host. We also describe a pro-pathogen role of macrophage-derived extracellular vesicles in

accession n° GSE243519 (https://www.ncbi.nlm.nih.gov/geo/query/acc.cgi?acc=GSE243519).

**Funding:** This study was financed in part by the Coordenação de Aperfeiçoamento de Pessoal de Nível Superior – Brasil (CAPES) – Finance Code 001 (sandwich PhD scholarship for RJAdeC) and the Conselho Nacional de Desenvolvimento Científico e Tecnológico – Brasil (CNPq) (PhD scholarship for RJAdeC). AA was or is currently supported by an Agence Nationale de la Recherche (ANR) grant (19AMRB000503, 20CE350007). The funders had no role in study design, data collection and analysis, decision to publish, or preparation of the manuscript.

**Competing interests:** The authors have declared that no competing interests exist.

*C. neoformans* infection and reinforce the impact of non-lytic exocytosis and the macrophage profile on the pathophysiology of cryptococcosis.

## Author summary

Dormancy enables human opportunistic pathogens to persist in the host in a limited replicative state, establishing a latent infection. Upon immunosuppression, dormant cells can reactivate and resume growth, leading to an active infection. *Cryptococcus neoformans* is the causative agent of cryptococcosis, a deadly yeast infection characterized by a latent phase. Building upon previous evidence that macrophages serve as a cellular reservoir for inducing and hosting dormant *C. neoformans* cells, we demonstrated that these phagocytes not only support dormancy, but can also paradoxically promote reactivation. Specifically, non-activated and anti-inflammatory macrophages promote the reactivation of a subset of dormant cells by the release of extracellular vesicles, as well as by yeast expulsion through a phenomenon known as non-lytic exocytosis. Importantly, dormant *C. neoformans* infection maintains macrophages in a non-inflammatory state required for reactivation. Our findings establish a determining role of macrophages, which is dependent on macrophage phenotype, in the maintenance or reactivation of dormant *C. neoformans* infection. In addition, we have uncovered a pro-pathogen role of non-lytic exocytosis and extracellular vesicle release in facilitating the reactivation of this model dormant yeast. Our study contributes to the understanding of the pathophysiology of cryptococcosis and potentially other latent infections.

## Introduction

Under adverse conditions such as those encountered during host infection, a subpopulation of microorganisms can enter a low metabolic state known as dormancy. The induction of dormant cells enhances their chances of survival, albeit at the temporary expense of interrupting cellular reproduction [1]. Cells can reach such a high level of commitment to dormancy that they are no longer able to reactivate and regrow upon returning to favorable environment settings, such as routine media, unless exposed to specific stimuli. These cells have been described as the viable but non-culturable (VBNC) cells [2,3]. VBNC cells in bacteria may benefit from increased tolerance to antimicrobial drugs, stress conditions, and evasion of immune system surveillance [2,4,5]. The triggers for microbial dormancy induction and reactivation are diverse in nature (i.e., physical, chemical or biotical) and source. Reactivation triggers encompass various factors, including amino acids, vitamins, proteins, and other biomolecules endogenously synthesized by the organism or provided by its host [6–8]. Studies aimed at elucidating induction and reactivation mechanisms for the VBNC state may substantially contribute to understanding the pathophysiology of latent infections, including those caused by fungi, such as histoplasmosis, paracoccidioidomycosis and cryptococcosis [6,9,10].

*Cryptococcus neoformans* is the major etiologic agent of cryptococcosis, which causes about 112,000 deaths annually worldwide and poses *C. neoformans* as a major global health threat [11–13]. To date, *C. neoformans* is the only pathogenic yeast species described to acquire the VBNC state [14,15]. The fungus is ubiquitous in the environment, and challenges hosts in early childhood [16,17]. Primary infection occurs following inhalation of desiccated yeast cells or basidiospores, which readily germinate into the yeast form in the host's lung [18,19]. The

yeast cells can either be cleared or contained by an asymptomatic pulmonary granulomatous response in immunocompetent hosts [20,21], persisting within macrophages and/or giant multinucleated cells [22,23]. Epidemiological [24,25] and serological [26] evidence supports that this latent infection lasts for months to decades before reactivation and clinical manifestation, mostly in immunocompromised patients (HIV infection or immunosuppressive treatment such as steroids therapy) [25–28]. Latent *C. neoformans* infection is postulated to be the manifestation of the dormant state acquired by the immunologically restrained fungus [9,15,29]. Conversely, the establishment of an active infection by *C. neoformans* comprises uncontrolled intracellular/extracellular replication of the yeast, as well as its dissemination from the lungs [22,30]. This dissemination process is aided by circulating monocytes, which act as carriers, and often leads to the deadly cryptococcal meningoencephalitis [31,32].

Monocytes and macrophages play paradoxical roles in cryptococcal infection, which are canonically determined by the nature of the T-helper cell response (whether Th1 or Th2 subtype) [33]. Under sensing of Th1-signaling cytokines (e.g., interferon gamma [IFN-γ]) an anticryptococcal macrophage response predominates. This response is marked by the intraphagolysosomal production of reactive oxygen and nitrogen species, typical of classically activated macrophages (M1 cells). Conversely, in the presence of Th2 signature cytokines, such as interleukin 4 (IL-4), the cell develop a phagolysosome environment more permissive for yeast survival and growth, characteristic of alternatively activated macrophages (M2 cells) [34–36]. These polarization states are dynamic and interchangeable [36,37]. During the course of infection, their respective macrophage populations coexist and undergo fluctuations alongside other phagocytes involved in the response against *C. neoformans* [29,37,38]. In addition to cytokines, many other factors secreted by both the yeast and macrophages can influence M1/M2 polarization and disease outcome. Among these factors are extracellular vesicles (EVs), nanosized membrane-limited structures containing complex cargo that can stimulate a protective M1 profile [39–41].

Despite its close association with the phagolysosome of host macrophages, *C. neoformans* is a facultative intracellular organism. Remarkably, *C. neoformans* is able to egress the phagocyte, especially M1 cells, leaving it unharmed, by a phenomenon known as non-lytic exocytosis or vomocytosis [30,34,42,43]. The adaptation of *C. neoformans* to the intracellular environment and its ability to undergo non-lytic exocytosis are survival strategies mediated by the production of diverse virulence factors such as urease and capsule [42,44,45]. Moreover, intracellular fitness is known to be involved in the induction of dormant yeast cells [46].

Previously, our team [14] experimentally replicated phagolysosome conditions such as nutritional deprivation, hypoxia, and acidic pH to generate a high yield of *C. neoformans* VBNC cells *in vitro*. These cells resemble the dormant yeast cell subpopulation induced during infection of mice and macrophage-like cells [46], paving the way for in-depth studies on the biology of VBNC cells and their interaction with host cells. VBNC cells exhibit stress-related transcriptional and posttranscriptional regulatory mechanisms, growth latency, and partial reactivation upon the addition of pantothenic acid (PA) [14,47], a vitamin involved in quorum sensing and synthesis of the central metabolic intermediate acetyl-CoA [48,49]. Nevertheless, research on the interplay between VBNC cells and macrophages is still warranted to better understand the mechanisms underlying the development of latent cryptococcal infection and its reactivation at the cellular level [15].

To address this issue, we first evaluated the transcriptomic response of bone marrow-derived murine macrophages (BMDMs) challenged with *C. neoformans* in different physiological states, including VBNC induced *in vitro* [14]. Afterwards, we validated at the translational level relevant host immune pathways regulated during *C. neoformans* infection. Furthermore, when assessing the impact of the macrophage activation profile on the outcome of

BMDM-VBNC cells interactions, we found that intracellular persistence maintains dormancy irrespective of the macrophage polarization state and the presence of known reactivation-promoting factors within the yeast-containing phagolysosome. Conversely, M0 and M2 macrophages facilitate reactivation by non-lytic exocytosis and the secretion of extracellular vesicles. Our results provide insight into the maintenance and awakening mechanisms of dormant *C. neoformans* underlying latent cryptococcal infection and its reactivation.

## Results

### VBNC cells elicit a reduced and specific transcriptional response in BMDMs

To explore the transcriptional landscape of host macrophage upon the interplay with dormant *C. neoformans*, we carried out a high-throughput RNA sequencing analysis (RNAseq) of resting BMDMs at six hours post-infection (hpi) with *C. neoformans* yeast cells in distinct physiological states: logarithmic phase (LOG), stationary phase (STAT) or the VBNC state. This early time point was chosen to avoid supernumerary proliferation of *C. neoformans* in the phagolysosome of BMDMs.

Principal component analysis (PCA) of the global transcriptome data and the heatmap of the differentially expressed genes (DEGs) showed specific patterns for each condition. VBNC elicited the closest transcriptional profile to that of uninfected control BMDMs (UNI) and were more similar to STAT than to LOG group (Fig 1A and 1B). Consistently, VBNC moderately perturbed host transcriptome by modulating 813 macrophage genes, which represents less than half of DEGs found in STAT-infected BMDM (n = 1991) and almost a third regulated by LOG (n = 2281). Out of the 813 DEGs, 45 (5.5%) were specifically regulated in the VBNC-BMDM co-culture. A total of 527 genes (64.8%) were shared among all conditions, while a set of 202 (24.8%) overlapped exclusively between VBNC and STAT and 39 (4.8%) between VBNC and LOG (Fig 1C).

To gain better biological insight into these DEGs, we conducted a Gene Ontology (GO) enrichment analysis of biological processes (S1 Table). The top 20 strongest p-value GO terms detected in each condition were depicted in Fig 2. DEGs in the VBNC condition clustered in fewer GO terms (n = 10) as compared to STAT and LOG conditions (n = 27 and 66, respectively). Out of the ten GO terms enriched in VBNC-infected BMDMs, six were unique to this condition, one of which is composed of genes linked to sequestering of triglyceride, the major storage form of fatty acids, whose metabolic pathways are critical for VBNC *C. neoformans* [14]. Other processes of particular relevance included macrophage chemotaxis and positive regulation of the nitric-oxide synthase biosynthetic process.

Furthermore, only genes related to inflammatory response and positively related to the extracellular signal-regulated kinase 1/2 (ERK1/2) cascade were common to all experimental groups. In particular, VBNC presented only 54 DEGs belonging to the former process, compared to 113 and 135 in STAT and LOG conditions, respectively. In agreement, DEGs modulated by STAT and LOG but not VBNC cells were categorized to positive regulation of inflammatory response, cell migration, and neutrophil chemotaxis processes, as well as cellular response to interleukin-1, interferon-gamma, and lipopolysaccharide processes. Other genes enriched exclusively by STAT and LOG included those associated with defense response to virus, intracellular protein transport, small GTPase mediated signal transduction, as well as positive regulation of GTPase activity, protein kinase B signaling, angiogenesis, and proteasomal ubiquitin-dependent protein catabolic processes. Put together, VBNC cells promotes a reduced and specific transcriptional response in BMDMs.

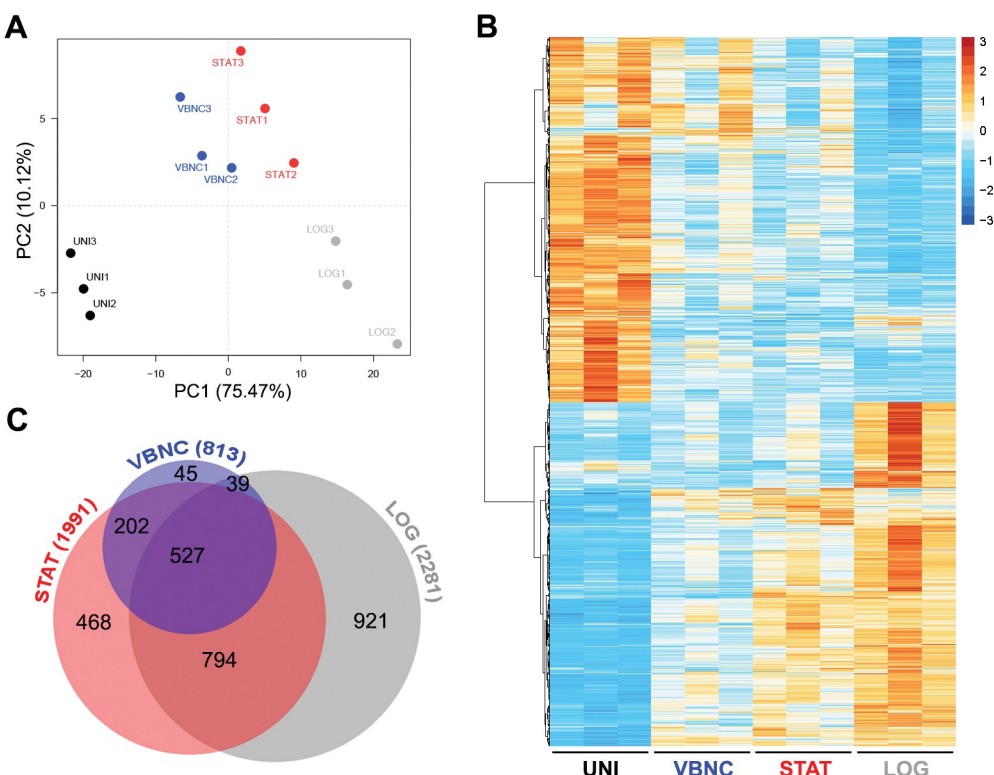

**Fig 1. VBNC cells promote a reduced and specific transcriptional response in murine bone marrow-derived macrophages (BMDMs).** Resting BMDMs were incubated with the opsonizing mAb 18B7 alone (UNI, uninfected control) or *C. neoformans* yeast cells in distinct physiological states: logarithmic phase (LOG), stationary phase (STAT) or VBNC cells. After 2 h, free unbound yeast cells and antibodies were washed, and the co-incubation protracted up to 6 h for RNA extraction. Principal component analysis (PCA) of the global transcriptome data (**A**) and heatmap of the differentially expressed genes (DEGs) (**B**) showed different patterns for each condition, with VBNC cells eliciting a specific pattern of modulation on macrophages, but close to STAT cells. The color gradient depicts normalized transformed counts. (**C**) Area-proporcional Venn diagram of DEGs showed that VBNC cells triggered the weakest transcriptional response, followed by STAT and LOG cells. Data from biological triplicates were used (adjusted p-values <0.05 and |Log2FC|>0.5).

## VBNC cells exert a low immunostimulatory activity in BMDMs

Given that VBNC cells triggered genes related to inflammatory response and macrophage chemotaxis (Fig 2), we deepened the analysis of the immunostimulatory capacity of VBNC cells. Cytokines, key inflammatory mediators, were used as readout. The RNAseq revealed that LOG cells induced a series of genes belonging to the CC and CXC chemokine families, mainly associated with the recruitment of monocytes and neutrophils, respectively. On the other hand, BMDMs infected with VBNC or STAT exhibited up-regulation of a reduced set of CC genes, while none of the CXC genes were up-regulated (Fig 3A). In addition, unlike LOG and STAT, VBNC cells did not induce the expression of some components of membrane-bound pattern-recognition receptors (PRRs) signaling pathways involved with cytokine production in response to fungal infections (S1 Fig) [50].

To validate RNAseq results, we prolonged co-cultures up to 24 h and screened a panel of cytokines in the cell supernatant by Luminex assay (Fig 3B and S2 Table). Consistently, VBNC cells maintained the low immunostimulatory capacity already known for *C. neoformans* in other physiological states (LOG or STAT) for most of the cytokines tested [51]. As an exception, CCL4, which is a chemoattractant for monocytes/macrophages, has a 2-fold increase in its expression. In line with the RNAseq results (Fig 3A), CCL2 secretion was also triggered by

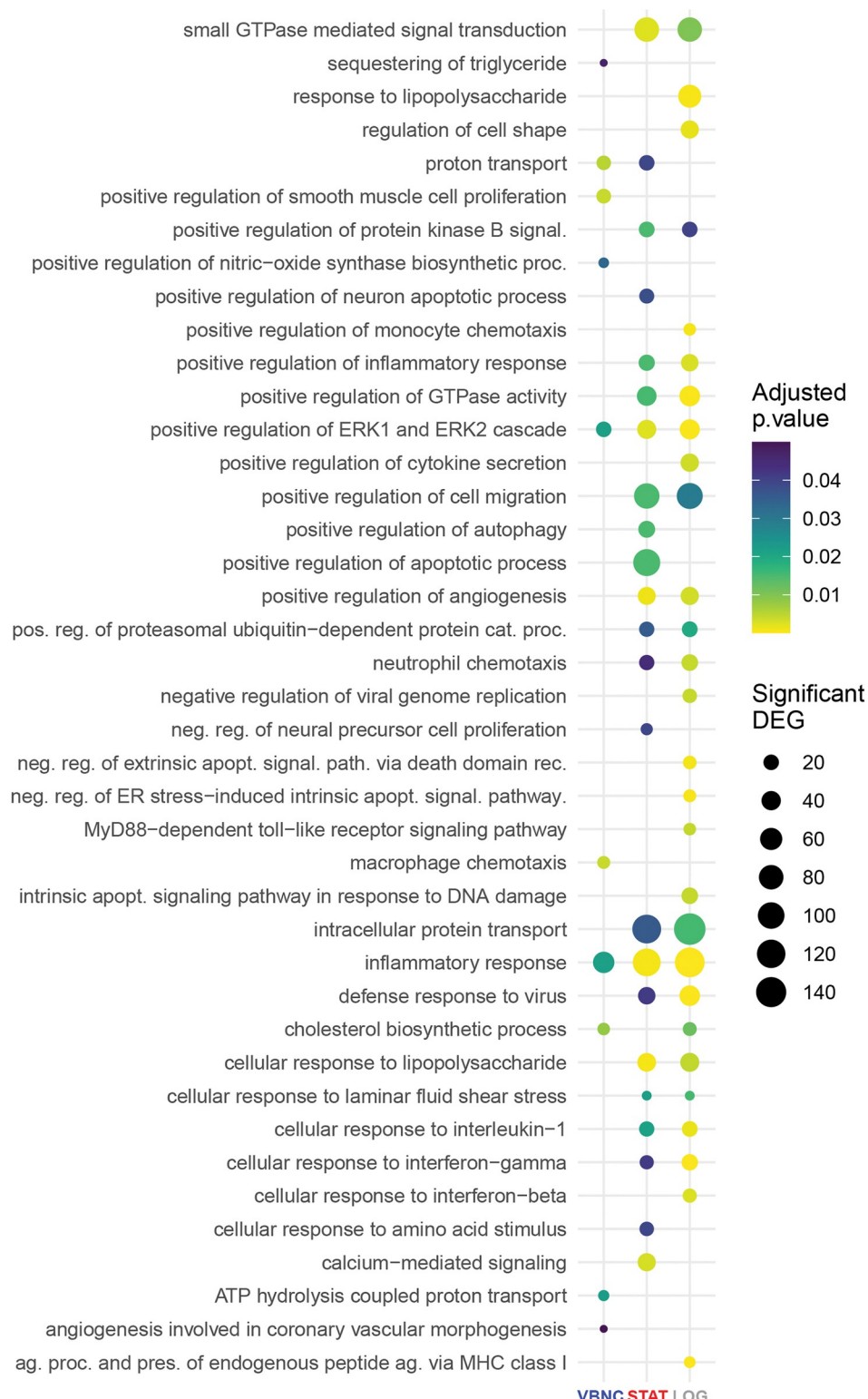

**Fig 2. Gene Ontology (GO) enrichment analysis of biological processes for the DEGs in BMDMs upon *C. neoformans* infection.** The top 20 strongest p-value GO terms detected in each condition are plotted. Dots are color-coded according to p-values, whereas the dot size is proportional to the number of significant DEGs (adjusted p-values <0.05) falling in the given GO term for each condition. Data from biological triplicates were used.

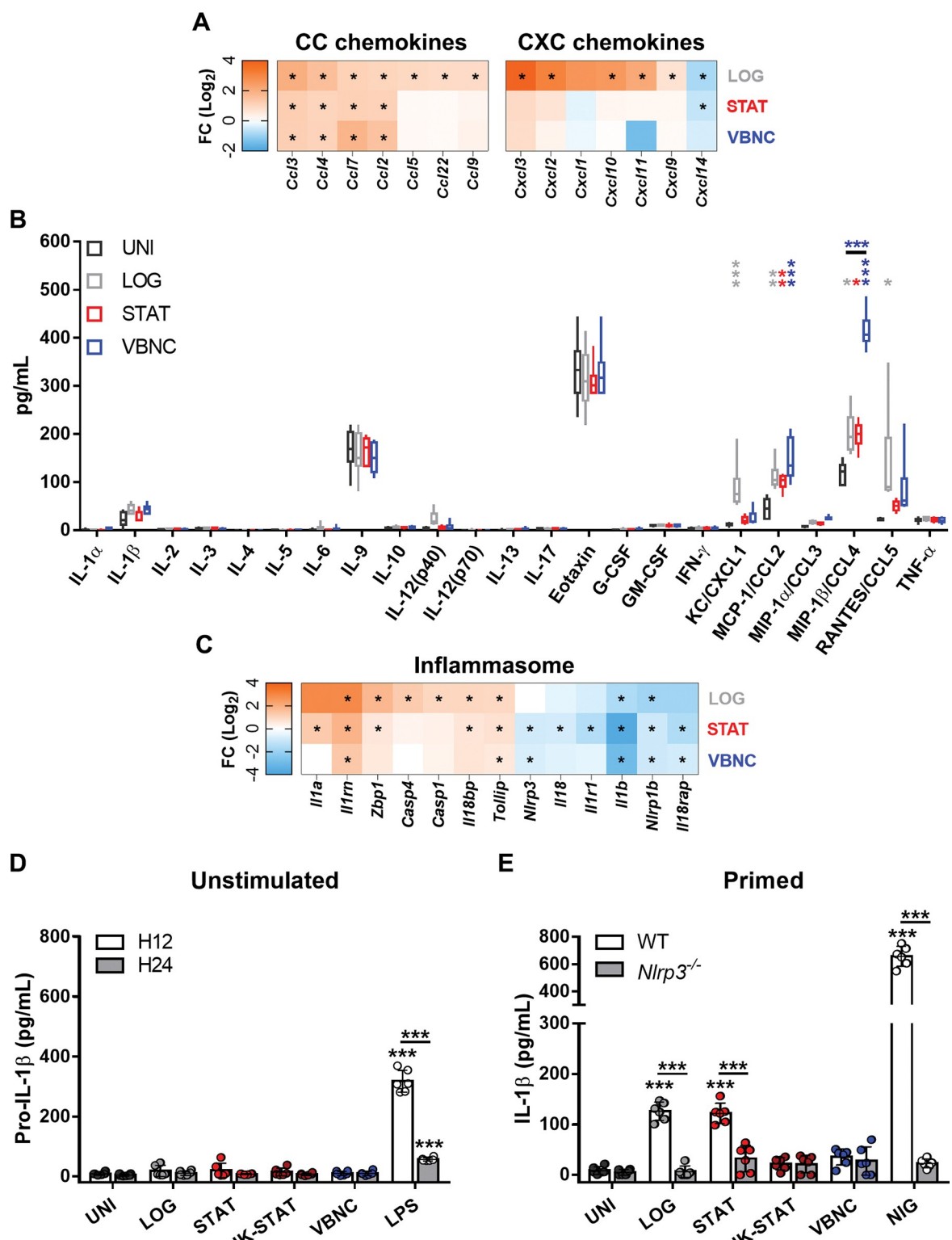

**Fig 3. VBNC cells exert a low immunostimulatory activity in BMDMs.** (**A**) Heatmap of CC and CXC chemokine genes differentially expressed upon *C. neoformans* infection. (**B**) Luminex screening of cytokines/chemokines in the cell supernatant after 24 h of interaction. (**C**) Heatmap of genes related to inflammasome signaling pathways showing inhibition under *C. neoformans* infection, especially by STAT and VBNC cells. (**D**) Intracellular pro-IL-1β levels detected by ELISA were unchanged during co-cultivation with *C. neoformans* for 12 h, while increased by stimulation with 500 ng/mL LPS (positive control). (**E**) VBNC cells do not activate the NLRP3 inflammasome. Increased IL-1β levels in the supernatant after 24 h of incubation of LOG, STAT cells or nigericin (NIG; 20 μM during the last 40 min, as a

positive control) with LPS-primed BMDMs derived from wild-type, but not NLRP3-knockout mice. No increase was detected under incubation with VBNC or STAT cells heat-killed at 70˚C for 1 h (HK-STAT). Data from biological triplicates (A and C) or duplicates (B, D and E) were used. Results represent the mean values ± standard deviations (SD) (bars) where each dot depict 1 technical replicate; or represent the median and interquartile range (box plot). Heatmaps of transcriptional profiles were generated using the average expression value of the biological triplicates; statistically significant (p<0.05) gene regulation is highlighted with an "*". *p<0.05, **p<0.01, ***p<0.001, between indicated groups or compared to uninfected control (UNI).

all *C. neoformans* physiological states, while CXCL1 and CCL5 secretion were induced only by LOG cells (Fig 3B). Furthermore, TNF-α was the sole non-chemokine cytokine tested that displayed transcriptional induction upon *C. neoformans* infection, regardless of the yeast cell state (S1 Fig), although it was not translationally induced (Fig 3B).

## VBNC cells are unable to activate the NLRP3 inflammasome

Interleukin-1 (IL-1) family cytokines are protective against *C. neoformans* infection [52–54]. Thus, we evaluated whether inflammasome signaling pathways, which are critically involved in the regulation of IL-1 family members, are differently affected by the challenge of BMDMs with VBNC cells. RNAseq analysis showed differential expression of a series of inflammasome genes in the macrophage upon infection with *C. neoformans* (Fig 3C). Overall, genes related to the sensing function of the inflammasome complex (*Nlrp3* and *Nlrp1b*), or recognition of its products (*Il1r1* and *Il18rap*) were downregulated. Accordingly, negative regulators of the IL-1 receptor signaling (*Il1rn* and *Tollip*) were induced. Consistently, *Il1b* was downregulated in all conditions. Likewise, we observed a trend towards *Il18* downregulation, as well as the upregulation of *Il18bp* (an IL-18 antagonist), *Il1a* and *Zbp1* (gene encoding an inflammasome sensor) under macrophage interaction with non-dormant forms. In addition, genes related to inflammatory caspases, Casp1 and Casp4, the former being the main NLRP3-inflammasome effector, were upregulated in LOG but not in STAT and VBNC conditions.

NLRP3 inflammasome activation classically occurs through two signals: (i) recognition of pathogen-associated molecular patterns (PAMPs), such as bacterial lipopolysaccharide (LPS), leading to the transcription and expression of pro-IL-1β protein (priming step); and (ii) subsequent recognition of danger-associated molecular patterns (DAMPs), resulting in the release of mature IL-1β [55]. Regardless of the condition, infection with *C. neoformans* did not affect pro-IL-1β levels after 12 or 24 hpi (Fig 3D) as assessed by ELISA. In contrast, LPS stimulation increased it significantly, as already described [56]. These results were consistent with the findings from our Luminex screening and RNAseq analysis (Fig 3B and 3C), suggesting that *C. neoformans* is unable to elicit inflammasome priming.

In accordance with our previous data [57], the interaction of LPS-primed BMDMs with LOG or STAT cells resulted in increased secretion of IL-1β at 24 hpi (Fig 3E). When we used LPS-primed BMDMs derived from NLRP3-deficient mice, IL-1β levels remained low at baseline, thus confirming an NLRP3 inflammasome-dependent secretion. Notably, both heat-killed STAT (HK-STAT; used as a control of fungal metabolism) or VBNC cells were unable to induce IL-1β secretion, regardless of BMDM priming with LPS or the mouse lineage from which BMDMs were obtained. Altogether, our results demonstrate that VBNC cells not only keep the ability of non-dormant yeast cells to suppress inflammasome priming but are also incapable of activating the NLRP3 inflammasome in macrophages.

## The cell wall composition of VBNC cells is consistent with their low immunostimulatory profile

Hypoxia is known to promote immunologically relevant remodeling of the fungal cell wall, as shown for pathogenic fungi [58,59]. Since we generate VBNC cells using starvation and

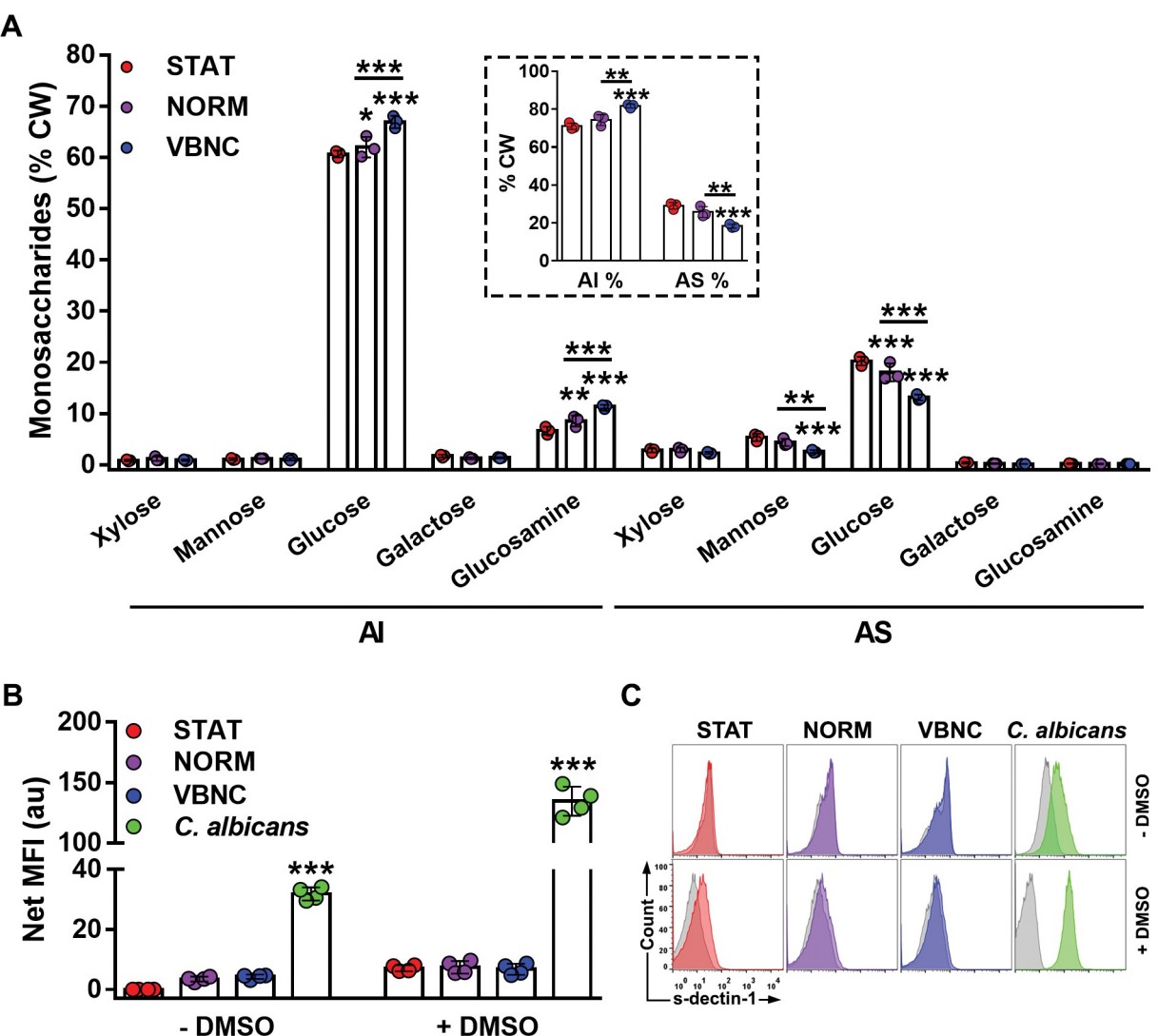

**Fig 4. The cell wall (CW) composition of VBNC cells is not drastically altered.** (**A**) Monosaccharide composition (% of CW total sugars) of the alkali-insoluble (AI) and alkali-soluble (AS) fractions of STAT; VBNC cells or its counterpart normoxic control group (NORM). *C. neoformans* cells present a higher percentage (of total CW dry weight) of AI than AS fraction, especially VBNC cells (insert). Data are means ± SD from biological triplicates analyzed by gas-chromatography. (**B**) Yeast cells were sequentially incubated with soluble Fc-conjugated dectin-1 (s-dectin-1) and Fc-specific IgG-FITC, and then acquired by flow cytometry. β-glucan exposure on the CW surface is similar among *C. neoformans* groups, with or without capsule removal with DMSO, while enhanced in *C. albicans* yeast cells (positive control). Data are means of median fluorescence intensity (FMI; arbitrary units) of cells labeled for β-glucan exposure subtracted of the MFI of its respective unlabeled control cells (net MFI) ± SD from biological duplicates. (**C**) Flow cytometry histogram profiles of s-dectin-1-labeled yeast cells (colored) versus unlabeled cells (grey) from 1 representative experiment of 2. *p<0.05, **p<0.01, ***p<0.001, between indicated groups or compared to STAT group.

hypoxia, we evaluated the cell wall composition of VBNC in parallel to that of a normoxic control (NORM), in which VBNC cells are also generated but to a lower extent [14]. Overall, the proportion of alkali-insoluble (AI, comprised of β-glucan, mainly with β-1,3-links, and chitin) to the alkali-soluble (AS, composed of α-glucan) [60] fraction significantly increased in the VBNC state as compared to the other conditions (Fig 4A). Furthermore, changes in cell wall composition were observed: (i) an increase of glucose and glucosamine in the AI-fraction in VBNC and to a lesser extent in NORM, reflecting an increase of β-glucan and chitin content, respectively; (ii) decrease in glucose and mannose in the AS-fraction in VBNC and NORM,

reflecting a decrease in the α-glucan levels (Fig 4A). Altogether, these findings indicate that the compositional variation detected is likely characteristic of the VBNC state rather than specific to the hypoxia stress response.

To verify whether these minor alterations could reverberate on β-glucan exposure, we used soluble dectin-1 receptor (s-dectin-1) to assess the recognition of β-glucan on cryptococcal cell wall. In contrast to the significant labeling observed for *Candida albicans*, which was used as a positive control, no increase in β-glucan exposure was noticed in STAT, as already described [61], nor in VBNC or NORM cells (Fig 4B and 4C). The removal of the capsule by DMSO treatment resulted in a slight but not significant increase in s-dectin-1 binding (Fig 4B and 4C), in agreement with the analysis of cell wall composition (Fig 4A). Of note, DMSO-treated *C. albicans* showed enhanced labeling, suggesting a positive effect of DMSO on the accessibility of the β-glucan within the cell wall. These results indicate that the cell wall of VBNC cells is not drastically different from that of STAT cells in terms of composition, structure, and immunostimulatory potential, in accordance with macrophage immune response assays.

## VBNC cells neither induce the production of nitric oxide nor neutralize it

The inducible nitric oxide synthase (INOS) enzyme responds to *C. neoformans* infection by catalyzing the hydrolysis of arginine into nitric oxide (NO), thus enabling the macrophage to control the growth and eliminate the yeast [36,62]. In this study, we observed that Slc7a1 and Slc7a2 genes, the main transporters for arginine acquisition [63], were positively regulated in LOG and STAT conditions, but not in VBNC as observed in the RNAseq analysis (Fig 5A and 5B). In addition, all physiological states of *C. neoformans* induced an alternative pathway for arginine obtention by the recycling of citrulline, which is produced concomitantly with NO by INOS, and converted back into arginine by enzymes encoded by *Ass1* and *Asl* genes. Accordingly, LOG and, to a lesser extent, STAT cells were able to induce iNOS (*Nos2*) expression, while VBNC cells could not. Expression of the gene coding for arginase (Arg1), an enzyme that competes with iNOS for using arginine as a substrate, was significantly repressed in STAT and VBNC conditions, with the same trend being observed for LOG infection although not significant.

Furthermore, we took a closer look at those genes positively associated with nitric-oxide synthase biosynthetic processing (GO:0051770) enriched only in VBNC condition (Fig 2 and S1 Table). VBNC cells showed a very similar modulatory pattern to STAT, as both upregulated *Kdr*, *Nampt*, *Ccl2*, and *Cpeb1*, the first two significantly less induced in VBNC compared to LOG condition; whereas down-regulated *Map2k6* and *Tlr9* (S2 Fig). LOG cells significantly modulated these same set of genes, and in addition *Stat1* and *Tlr2* (both transcriptionally enhanced), but not *Cpeb1* and *Map2k6*.

We further evaluated the levels of nitrite ($NO_2^-$), as a proxy for NO production, in the cell supernatant after 24 h of interaction. None of the physiological states of *C. neoformans* led to an effective nitrite production (Fig 5C). Considering that *C. neoformans* is capable of detoxifying NO, we evaluated its indicator levels in cultures of infected macrophages stimulated with IFN-γ and LPS, known as strong NO inducers [52]. Interestingly, while STAT and LOG reduced nitrite levels induced by IFN-γ and LPS, interaction with HK-STAT or VBNC resulted in unchanged levels of the NO by-product (Fig 5D). Taken together, our results suggest that VBNC cells do not stimulate nitric oxide production during macrophage infection and fails to neutralize nitric oxide upon interaction with activated macrophages.

## Intracellular VBNC cells maintain phenotype and viability within BMDMs

Since VBNC cells are poor inducers of the immune response, we wondered if the macrophage polarization phenotype could provide favorable or detrimental conditions for the reactivation

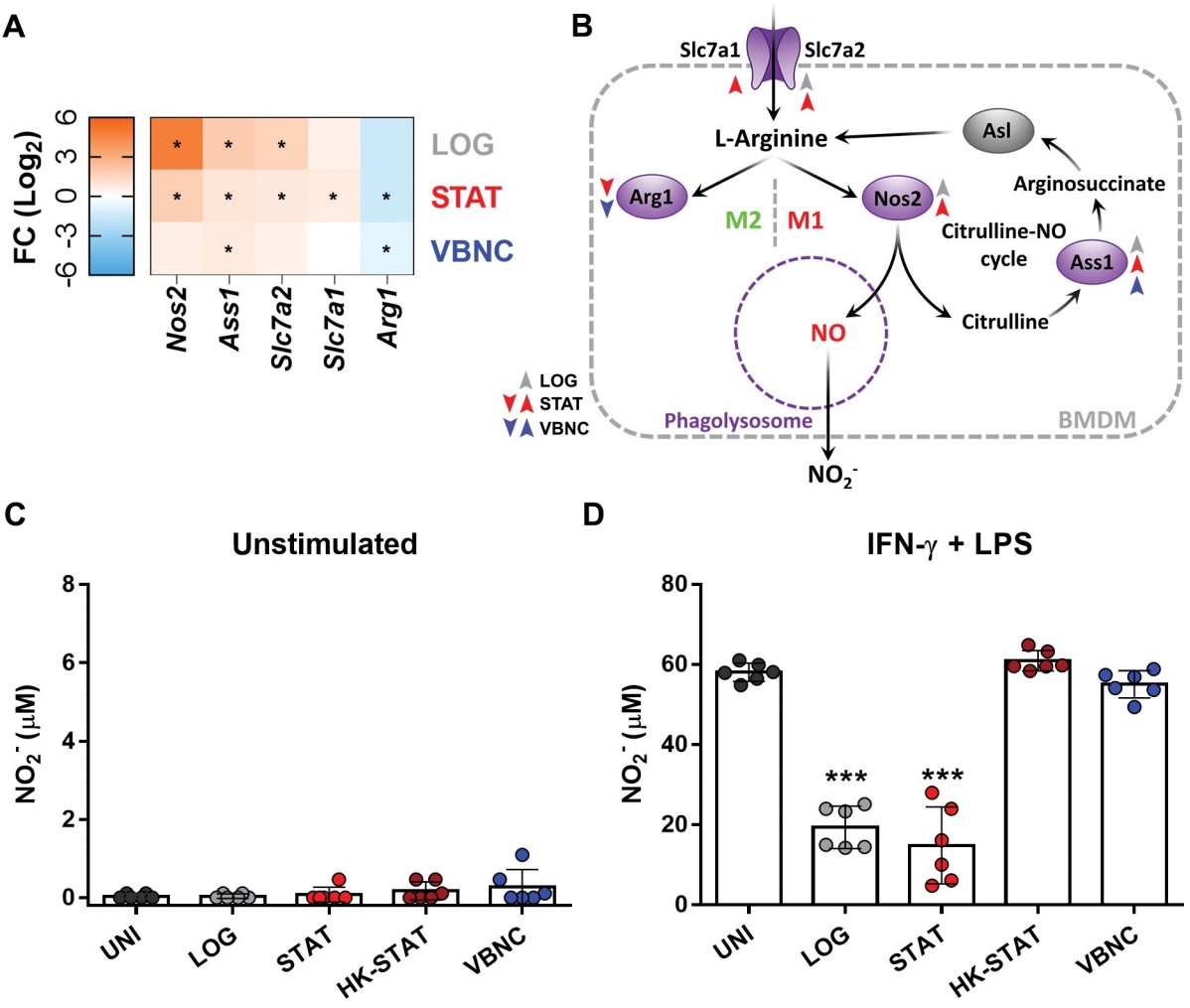

**Fig 5. VBNC cells neither induce the production of nitric oxide (NO) nor neutralize it.** (**A**) Heatmap showing induction of genes related to arginine uptake and metabolization via inducible nitric oxide synthase (INOS/Nos2) during BMDM infection with LOG or STAT, but not VBNC cells. (**B**) Representative scheme of the arginine metabolic pathways: after uptake by Slc7a1 and/or Slc7a2, L-arginine is predominantly metabolized by Arginase 1 (Arg1) in M2 cells; or by Nos2 in M1 cells, producing NO, which is spontaneously oxidized to nitrite ($NO_2^-$). Gene regulation by LOG, STAT or VBNC cells are represented by grey, red and blue arrowheads, respectively. Up and down arrowheads denote positively or negatively regulated genes, respectively. Asl gene (grey) is constitutive. Indirect measurement of NO production by assessment of nitrite from the supernatant of BMDMs unstimulated (**C**) or stimulated (**D**), with the NO inducers LPS and IFN-γ (500 ng/mL and 500 IU, respectively), by Griess reaction method. BMDMs were first incubated for 2 h with opsonized *C. neoformans* cells, washed to eliminate unbound yeast cells, and then added of stimuli for NO production for additional 22 h. STAT cells heat-killed at 70°C for 1 h (HK-STAT) were used as control. Data are means ± SD from biological duplicates where each dot depict 1 technical replicate. Heatmaps of transcriptional profiles were generated using the average expression value of biological triplicates; statistically significant (p<0.05) gene regulation is marked with an "*". ***p<0.001, compared to uninfected group (UNI).

of VBNC cells. More precisely, to acquire different macrophage activation profiles, we treated resting BMDMs (M0; non-activated status) with major inducing cytokines for the classic (M1) or alternative (M2) activation states (IFN-γ and IL-4, respectively), based on Murray et al. [64] (Fig 6A). Treatments with LPS (combined with IFN-γ) or the corticosteroid dexamethasone (Dex) were applied to ensure the acquisition of M1 or M0 states, respectively. Signature markers for M1 and M2 profiles were checked and confirmed the expected polarization phenotypes (Figs 6B and S3). Of note, stimulation with IFN-γ alone weakly induced an M1 phenotype, which was fully recapitulated upon co-stimulation with LPS (Figs 6B and S3).

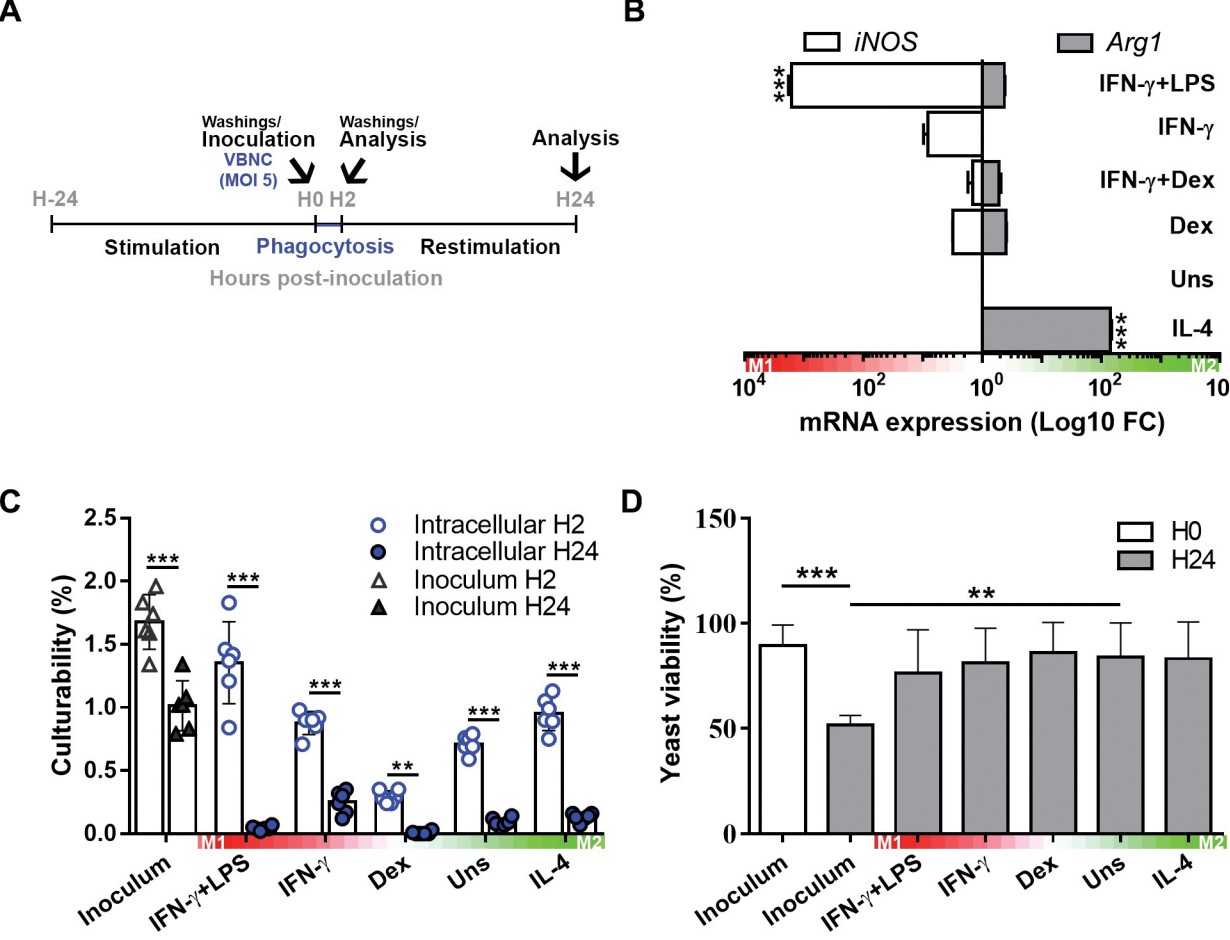

**Fig 6. Intracellular VBNC cells maintain phenotype and viability in BMDMs.** (**A**) Scheme depicting the cell treatments used to obtain BMDMs with different activation states. Prior to infection, resting BMDMs (non-activated; M0) were stimulated during 24 h towards the classic (M1) or alternative (M2) activation states with IFN-γ (500 IU) or IL-4 (20 ng/mL), respectively. LPS (100 ng/mL; concomitantly with IFN-γ) or dexamethasone (0.1 μM; Dex) were applied to ensure a M1 or M0 state, respectively. Stimuli were washed and the BMDMs inoculated with VBNC cells at a multiplicity of infection (MOI) of 5. As a control, yeast cells were plated in the absence of phagocytes (Inoculum). After 2 h of phagocytosis, unphagocytosed yeast cells were washed out and the BMDMs restimulated (LPS was increased to 500 ng/mL). (**B**) Before infection (hour 0; H0), the expression of signature markers for M1 (iNOS) and M2 states (Arg1) were checked using real-time PCR analysis. The gene expression was normalized to the constitutive control GAPDH. (**C**) The culturability of VBNC cells decreases over time upon internalization by BMDMs, irrespective of macrophage polarization phenotypes. After 2 and 24 h of interaction, internalized yeast cells were released using PBS containing 0.05% SDS and tested for culturability on sabouraud agar. Culturability data were expressed as means ± SD percentage of colony-forming unit (CFU) counts relative to the number of cells plated. Data from 1 representative experiment out of 2. Each dot depicts 1 technical replicate. (**D**) Yeast cells retrieved intracellularly after 24 h were also checked for viability by flow cytometry using the membrane-impermeant Live/Dead staining dye. Heatmaps represent the gradient of M1/M2 polarization. Data are means ± SD from biological duplicates. *p<0.05, **p<0.01, ***p<0.001, between the indicated groups or compared to unstimulated (Uns) or inoculum H24 group (infection assays).

We then co-cultured these macrophages with VBNC cells or kept the latter in media (RPMI + 10% fetal bovine serum [FBS]) alone as control (Inoculum) (Fig 6A). After 2 h of phagocytosis, non-internalized yeast cells were washed out, and intracellular yeast cells recovered for culturability analysis. In parallel, washed co-cultures were protracted up to 24 h with fresh medium containing stimuli to maintain macrophage polarization and processed for culturability and viability analysis of intracellular yeast cells.

Internalization of VBNC cells, regardless of the macrophage polarization state, led to a significant decline in fungal culturability, nearly reaching almost null levels at 24 h, as assessed by CFU compared to the inoculum control (Fig 6C). Similarly, yeast cells in the absence of

macrophages (inoculum control) in cell culture medium also decreased culturability over time but to a much lesser extent. Along with a loss in culturability, a partial reduction of viability was observed in this group (Fig 6D). In contrast, the viability of internalized yeast cells was unaltered, regardless of the hosting macrophage (Fig 6D). These results indicate that the internalization and persistence of VBNC cells within macrophages supports the maintenance of the phenotype and viability of VBNC.

## Macrophage phagolysosome inhibits reactivation despite containing reactivation-promoting factors

We postulated that internalized VBNC cells may not be able to reactivate because the phagolysosome lacked reactivation-promoting factors, such as pantothenic acid (PA), a key metabolic precursor of coenzyme A (CoA) [65]. The biosynthesis of this vitamin is catalyzed by pantothenate synthase, which is encoded by the CNAG_07414 gene in *C. neoformans*. To investigate this hypothesis, we used the growth of 07414Δ mutant strain as a proxy of PA levels within phagolysosomes. Indeed, 07414Δ lacks pantothenate synthase, making it auxotrophic for PA, as confirmed by seeding in minimal medium agar supplemented or not with this compound (Fig 7A). Supplementation of minimal medium with a range of primary/secondary acetyl-CoA precursors used by *C. neoformans* during macrophage infection [48,66,67] did not yield growth (S4 Fig), corroborating that PA (or its derivatives) is an essential precursor for CoA biosynthesis [68,69]. We determined that the minimum requirement of PA for growth of 07414Δ strain is 12.5 nM (green curve) (Fig 7B), and this is the same minimum concentration of PA observed to decrease latency of VBNC cells (Fig 7D and 7E) (i.e., promote reactivation), as previously determined [14]. In addition, by using the 07414Δ mutant, we indirectly identified the presence of pantothenic acid in the conditioned medium obtained from cultures of STAT cells (Fig 7C), as previously demonstrated directly [49]. However, we did not detect pantothenic acid in the conditioned medium derived from VBNC cells (Fig 7C). Taken together, these results suggest that VBNC cells have an impaired capacity to synthesize PA and, consequently, to reinitiate growth easily.

Finally, we co-incubated the 07414Δ strain or its parental wild-type strain (KN99α) with J774 cells (Fig 7F and 7G), known to be highly permissive for the intracellular proliferation of *C. neoformans* [70]. As a control, yeast was incubated alone in the presence of the cell culture medium DMEM (plus 10% FBS), which contains a high concentration of PA. STAT 07414Δ yeast cells were able to proliferate at levels comparable to KN99α not only in the extracellular medium containing PA, but also intracellularly (Fig 7F and 7G). These results were consistently reproduced when we conducted the experiment using BMDMs (S5 Fig). Since 07414Δ strain is auxotrophic for PA, this observation suggests that *C. neoformans* acquires PA or its derivatives from the macrophage phagolysosome. Indeed, *C. neoformans* possesses two genes, CNAG_00540 and CNAG_07402, which encode PA transporter proteins. These genes seem to functionally overlap with each other, as suggested by the enhanced growth observed in 00540Δ and 07402Δ deletion mutants in the presence of PA (S6 Fig).

Collectively, our results suggest that macrophage phagolysosomes supply *C. neoformans* with reactivation-promoting factors which may be PA or a mixture of PA and/or other factors yet to be characterized; however, other factors within the phagolysosome, which negatively affect reactivation, appear to predominate.

## Resting and M2-polarized macrophages allow reactivation of non-lytic-exocytosed VBNC cells

*C. neoformans* can escape the macrophage phagolysosome by non-lytic exocytosis [71], thereby returning to the extracellular matrix, a common histological finding in active

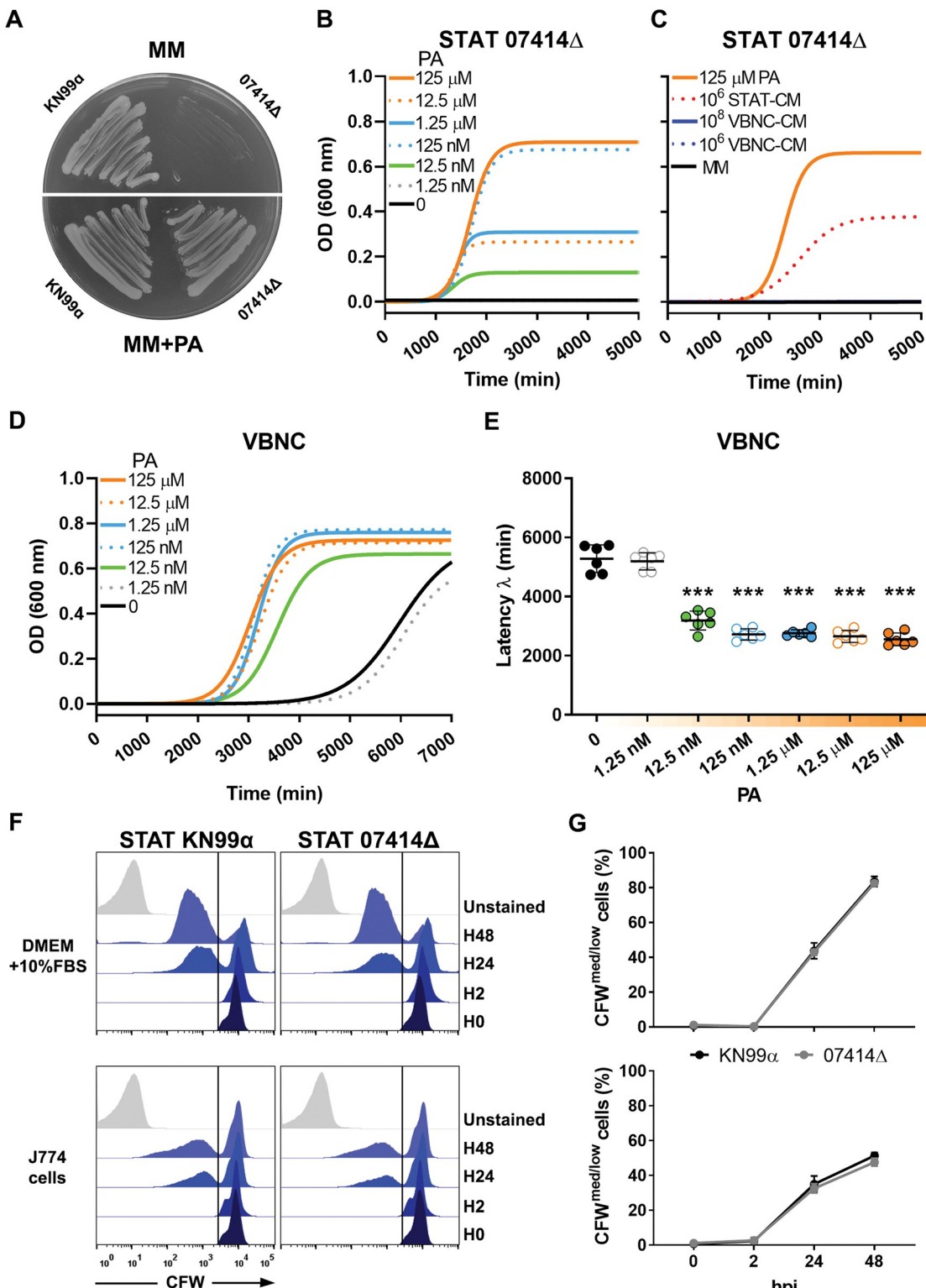

**Fig 7. Macrophage phagolysosome inhibits reactivation despite containing reactivation-promoting factors.** (**A**) *C. neoformans* 07414Δ mutant strain and its parental strain KN99α were spread on minimal medium (MM) agar supplemented or not with 125 μM pantothenic acid (PA). The 07414Δ mutant, which lacks pantothenate synthase, did not grow in the absence of PA, indicating that it is auxotrophic to this vitamin and therefore exploitable as a proxy for PA. (**B**) Growth curves of the 07414Δ mutant incubated in a base-10 serial dilution of PA in MM (from 1.25 nM to 125 μM PA) indicate that a concentration of 12.5 nM

PA (green curve) is the minimum required to support the growth of *C. neoformans*. (**C**) Growth curves of 07414Δ mutant incubated with 50% conditioned medium (v/v) derived from STAT (STAT-CM) or VBNC cells (VBNC-CM) cultured at $10^6$–$10^8$ cells/mL in MM for 4 h. In contrast to $10^6$ STAT-CM, the $10^6$–$10^8$ VBNC-CM was unable to support growth of the mutant, implying that VBNC cells demonstrate a diminished production of PA. Growth curves (**D**) and the derivative latency of growth (**E**) of *C. neoformans* H99O VBNC cells corroborate the findings of a previous report [14], which indicates that 12.5 nM PA is the lowest concentration of PA required to decrease latency in VBNC cells. (**F**) *C. neoformans* wild-type strain KN99α and 07414Δ mutant cells were stained with calcofluor white (CFW), opsonized, and incubated with LPS-primed J774 cells (MOI 2.5) or plated alone in DMEM added of 10% FBS (control rich in PA). Yeast collected from the experimental media alone or recovered after macrophage lysis at the indicated time points were acquired by flow cytometry and represented in flow cytometry histogram panels of cell counts vs. CFW signal. The unstained (H0) and HO (stained) control samples of each strain are depicted in both histograms of their respective strains. (**G**) The percentage of intracellular KN99α and 07414Δ cells harboring medium (med) or low CFW fluorescence (gated in panels), consistent with newly formed daughter cells, increases similarly over time, indicating that the macrophage phagolysosome contains PA/PA derivatives. Data are means ± SD from biological duplicates. Growth curves and fluorescence histograms are from 1 representative experiment of 2. ***p<0.001, compared to wells with no addition of PA.

cryptococcosis [22]. Given that, we evaluated if the VBNC cells recovered from the co-culture supernatant after 24 h of interaction with macrophages (Fig 8A) were more prone to reactivation. Remarkably, extracellular yeast cells experienced a significant increase (more than 4 times) in culturability, compared to the inoculum at 2 h, in a similar manner among almost all macrophage treatment conditions except LPS and IFN-γ induced BMDMs. Indeed, VBNC cells co-cultured with M1 macrophages activated with both LPS and IFN-γ showed a loss of culturability to levels similar to that of intracellular yeast cells (Fig 8A). These results indicate that interaction with resting or M2 macrophages promotes the extracellular reactivation of VBNC cells that exit macrophages. Conversely, reactivation is thwarted by M1 macrophages.

To confirm that the yeast cells retrieved from the cell-supernatant resulted from non-lytic exocytosis events, we implemented a flow cytometry-based assay described previously (Fig 8B and 8C) [57]. Yeast cells were labeled with Calcofluor White (CFW) to discriminate between parental (CFW$^{high}$) and daughter cells (CFW$^{med/low}$), which were gated out to avoid quantification of false-positive non-lytic exocytosis events related to yeast proliferation. Then, the labeled cells were incubated with unstimulated BMDMs. After the removal of non-internalized fungi after 2 h, extracellular (Fig 8B) and intracellular (Fig 8C) yeast cells were obtained at the indicated time points.

As expected, we observed an increase in the extracellular presence of parental STAT cells starting from 6 h (Fig 8B) and parental VBNC cells at 24 h. Consistently, a decrease was observed intracellularly at 24 h for STAT and VBNC groups (Fig 8C). Furthermore, heat-killed (HK) STAT cells alone or with the addition of 1% living STAT cells, used as controls, did not show a significative extracellular increase despite a significant intracellular decrease detected at 24 h.

Non-lytic exocytosis is known to be influenced by the intraphagolysosomal pH and enhanced by urease activity [44,71]. In line with this, we confirmed that VBNC cells exhibited fewer exocytosis events due to lack of urease activity by the Christinsen's urea broth test (Fig 8D). Taken together, our results indicate that resting and M2 macrophages allow reactivation of non-lytic-exocytosed VBNC *C. neoformans*.

## Macrophages promote reactivation by non-lytic exocytosis and release of EVs

We then sought to further analyze the mechanism underlying the extracellular reactivation of VBNC cells during interaction with non-inflammatory macrophages. We incubated VBNC cells (inoculum) under four conditions: (i) in medium alone; (ii) with BMDMs; (iii) with BMDMs but without 18B7 opsonizing mAb or (iv) with separation of the VBNC and BMDMs by a transwell system. The transwell insert used features a 0.4-μm semipermeable membrane,

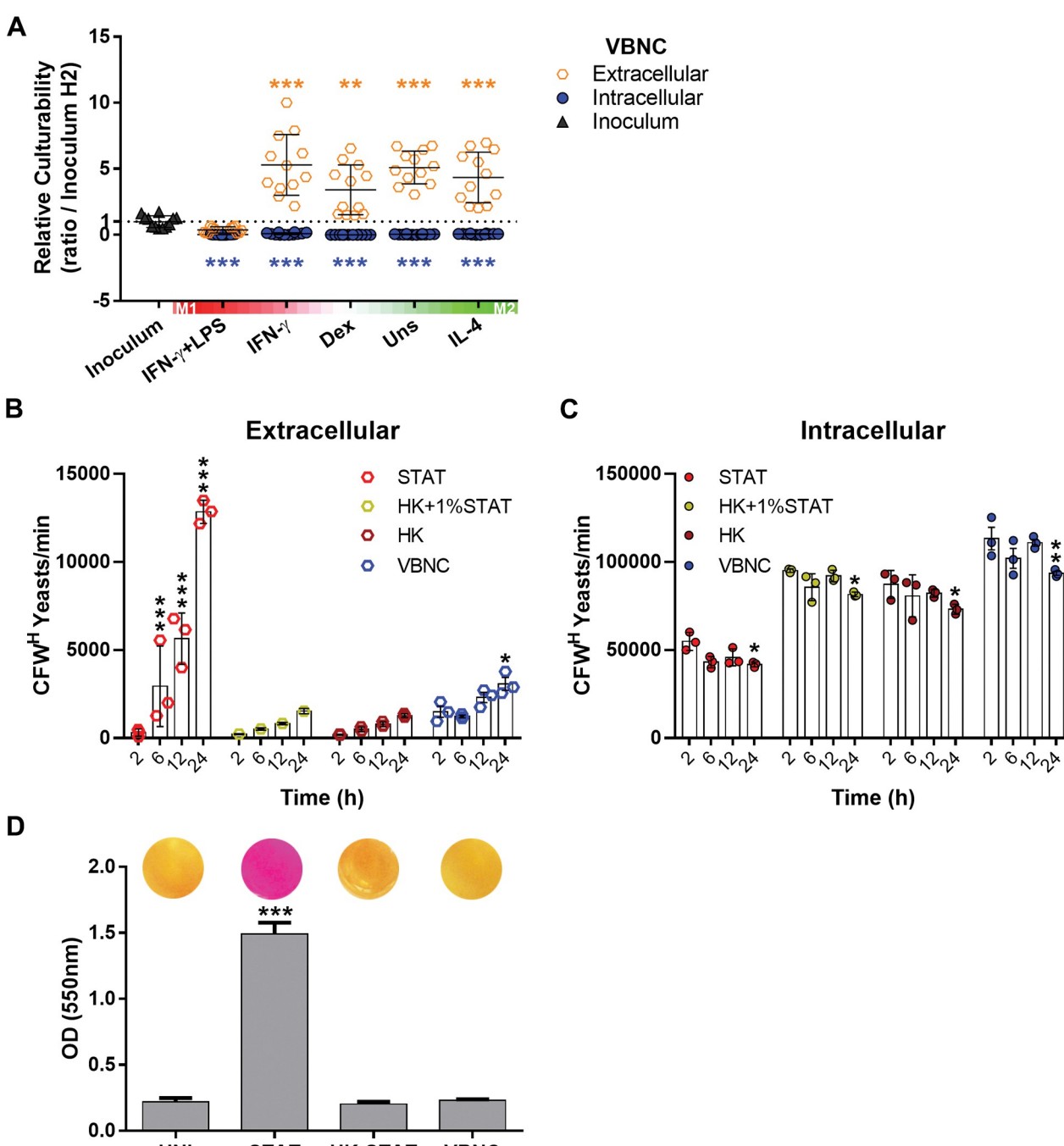

**Fig 8. Resting (M0) and M2 macrophages promote the reactivation of non-lytic-exocytosed VBNC cells.** (**A**) VBNC cells were incubated with BMDMs at different polarization states (indicated by the heatmaps) or plated in the absence of phagocytes (Inoculum). After removal of non-internalized fungi after 2 h, yeast cells found extracellularly after 24 h in the co-culture with M0, M2, and IFN-γ-polarized M1 cells exhibited enhanced culturability as determined by CFU assay. However, yeast cells co-cultured with IFN-γ-LPS-polarized M1 cells did not show an increased culturability. This experiment was conducted in parallel with the experiment shown in Fig 6C and shares the same inoculum and intracellular condition as controls. Relative culturability was calculated between the culturability of each group at 24 h time point and the inoculum group at 2 h. Ratios that crossed a threshold value of 1 (pointed line) were used as an indicator of reactivation. (**B** and **C**) STAT and, to a lesser extent, VBNC cells are exocytosed by BMDMs. Yeast cells were labeled with calcofluor white (CFW) and incubated with unstimulated BMDMs. After washing of non-internalized fungi after 2 h, extracellular (**B**) and intracellular (**C**) yeast cells were obtained at the indicated time points for enumeration of parental cells (CFW high; CFW$^H$) by flow cytometry. Control groups of heat-killed STAT cells (HK) alone or added of 1% STAT cells (HK+1%STAT) were included. (**D**) STAT but not VBNC cells show urease activity in Christensen's urea broth (2 h, 37°C, 100 rpm). A representative photography of culture wells is shown above bars. Data are means ± SD from biological duplicates where each dot depicts 1 technical replicate. The exocytosis assay data are from 1 representative experiment out of 2. *p<0.05, **p<0.01, ***p<0.001 compared to inoculum group (**A**), 2h time point (**B** and **C**) or uninoculated group (UNI) (**D**).

allowing unrestricted exchange of soluble factors (Fig 9A). To simplify our experimental approach, we employed M0 BMDMs as non-inflammatory macrophages instead of M2 BMDMs, as both cell types demonstrated an equal capacity to promote reactivation (Fig 8A).

All experimental conditions consisting of a co-culture with the M0 macrophages resulted in increased culturability of extracellular VBNC cells compared to the inoculum control (Fig 9B). The uppermost increase was observed in the condition allowing phagocytosis (Fig 9B). Thus, reactivation was positively influenced by but not dependent on internalization. Either direct or indirect contact between VBNC and BMDM cells yielded the same level of reactivation. This indicates that a soluble factor released by BMDMs, the secretion of which does not depend on cell-to-cell interaction, is responsible for the extracellular reactivation of VBNC cells.

EVs are nanosized carriers of a variety of bioactive compounds that play an important role in the host-pathogen interaction, including cryptococcal infection [41,72]. In light of this, we isolated EVs from M0 BMDM cultures and tested on VBNC cells in minimal medium to check the impact on its latency of growth (S7 Fig). Of note, the BMDMs were cultured using ultrafiltered FBS, which is depleted of EVs. Importantly, BMDM-derived EVs (10 μg/mL of protein content) were able to decrease growth latency of VBNC cells in comparable levels to those obtained with the addition of PA (125 μM) (S7 Fig). Afterwards, we explored the effect of macrophage polarization on the capacity of their respective EVs to reduce VBNC latency (activation of reactivation). We observed no significant differences among EVs derived from M0, M1, and M2 cells, indicating their similar activities (Fig 9C). Moreover, we examined EVs isolated from M0 peritoneal macrophages (PMs), which are representative of tissue terminally-differentiated primary macrophages, and from the human monocytic cell line THP-1. While we observed that their effect was not as pronounced as the PA control or EVs produced by BMDMs, irrespective of the BMDM polarization state, both PMs and THP-1 cells produced EVs leading to a significant decrease in latency, similarly to each other (Fig 9C). Collectively, our results indicate that macrophages can promote the reactivation of VBNC cells by non-lytic exocytosis and release of EVs (Fig 9D).

## Discussion

Macrophages are closely associated with *C. neoformans* cells during infection and have been shown to strikingly impact the outcome of cryptococcosis in paradoxical ways depending on their polarization state [33,34,36,73]. Hence, these highly plastic phagocytes represent a key link between latent cryptococcal infection and reactivation and host immune response. Here, we report that macrophages play a dual role in the control of *C. neoformans* reactivation from dormancy. Macrophages can sustain intracellular parasitism, leading to dormancy. Conversely, macrophages can also facilitate the reactivation of dormant cells through fungal expulsion by non-lytic exocytosis and the release of EVs (depicted in Fig 9D). Furthermore, reactivation is determined by macrophage polarization in response to cytokines and thus is likely favored by the capacity of *C. neoformans* to maintain its low immunostimulatory profile during dormancy. Previously, Hommel et al. [14] described an *in vitro* protocol to generate VBNC *C. neoformans*, enabling studies on the interaction between dormant yeast and host cells. This study demonstrated a reduced and unique modulation of the yeast transcriptome in the VBNC state, diverging from the stationary and, mainly, logarithm phase of growth.

Herein, we wondered if VBNC cells would differently modulate murine BMDMs, providing cues to understanding macrophage behavior during latent infection. Overall, our results are align with previous transcriptomics showing that *C. neoformans* extensively alters the transcriptional landscape of macrophages and monocytes, especially regarding immune/inflammatory response [74–76]. Moreover, we observed a reflection of yeast physiological state on its

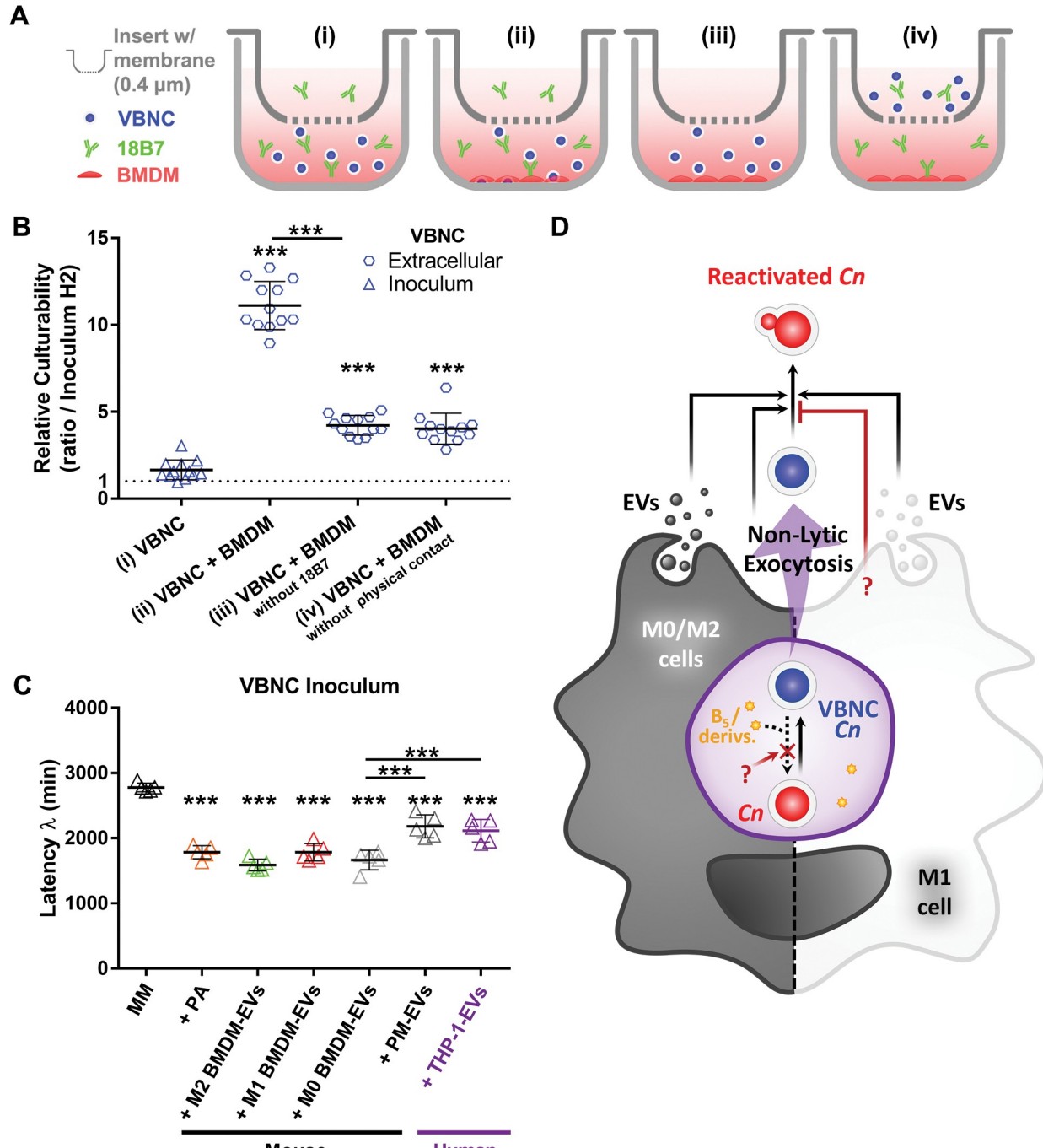

**Fig 9. Macrophages facilitate reactivation by the secretion of extracellular vesicles (EVs).** (**A**) Experimental design scheme of the transwell assay. VBNC cells were co-cultured with resting BMDMs under 4 conditions: (i) in medium alone; (ii) with BMDMs; or with BMDMs but precluded from phagocytosis by either (iii) removal of 18B7 opsonizing mAb or (iv) separation of inoculum and BMDMs from physical contact by a transwell system (0.4 μm pore size). The culturability of yeast cells recovered from the medium alone (inoculum) or the co-culture supernatant (extracellular) after 24 h of incubation was assessed by CFU assay. (**B**) Exposure to resting BMDMs secretome in the transwell assay enhanced culturability of VBNC cells. Relative culturability was calculated between the culturability of each group at 24 h time point and the inoculum group at 2 h (pointed line). (**C**) Latency of growth of VBNC cells incubated in minimal medium (MM) decreases upon the addition of pantothenic acid (PA; 125 μM) or EVs (10 μg/mL) isolated from cultures of diverse monocytes/macrophages. PM stands for Peritoneal Macrophages. Data are means ± SD from biological duplicates where each dot depicts 1 technical replicate. ***p<0.001, between the indicated groups or compared to control groups (VBNC or MM). (**D**) Proposed model for VBNC *C. neoformans* interaction with macrophages. Macrophages can promote induction and maintenance of the dormant state in *C. neoformans* (*Cn*) within the phagolysosome (shown in purple). This phenomenon occurs irrespective of the macrophage polarization phenotype and despite the presence of reactivation-promoting factors such

as vitamin B5 (B5; pantothenic acid) or its derivatives (derivs.). Conversely, macrophages can also facilitate reactivation. Specifically, M0 and M2 cells promote the extracellular reactivation of VBNC cells through the secretion of EVs and non-lytic exocytosis. In addition, isolated M1-derived EVs are equally capable of reactivating VBNC cells. However, M1 cells promote non-lytic exocytosis of VBNC cells without triggering extracellular reactivation, thereby prophylactically depleting a reservoir of dormant cells. It remains to be elucidated how the phagolysosome counteracts the capacity of B5/derivs. to reactivate intracellular VBNC cells and how M1 cells inhibit reactivation of extracellular VBNC cells (question marks).

capacity to impact BMDMs' transcriptome. Compared to non-dormant cells, especially logarithm-phase cells, VBNC cells modestly perturbed host transcriptome, regulating fewer genes, clustered in far fewer GO terms, most of them unique to this infection. VBNC cells clearly elicited a less pronounced transcriptional immune response: in contrast to non-dormant infection, it did not modulate groups of genes related to positive regulation of inflammatory response, neutrophil chemotaxis, and response to cytokines, LPS, and virus. Furthermore, in contrast to LOG cells, which elicited a robust neutrophil-recruitment response, VBNC and the quiescent phenotype STAT cells did not trigger upregulation or secretion of any neutrophilic chemokines, protective mediators against murine cryptococcosis [77,78]. In contrast to our findings, hypoxia-induced dormant *Mycobacterium tuberculosis* profoundly alters the transcriptome of Rhesus BMDMs, evoking a broad proinflammatory response to *M. tuberculosis* marked by TNF expression [79]. Other studies corroborate the correlation between a heightened proinflammatory response and the persistence of dormant *M. tuberculosis* in host granulomas [80–82]. Hence, different strategies for dormancy persistence may occur in the host.

Importantly, dormant *C. neoformans* retained the ability to provoke gene regulation of monocyte chemotaxis, leading to the secretion of CCL2, chemokine associated with cryptococcal granuloma formation [83,84]; and CCL4, which was particularly more secreted in this infection. Although CCL4 is increased in the lungs of various animal models [85,86] and in the human cerebrospinal fluid in response to *C. neoformans* infection [87], the role of this chemokine in cryptococcosis is still not well understood and appears not to be as critical as CCL2 [87–90]. Thus, we hypothesize that the ability of dormant *C. neoformans* to induce a specific and less pronounced inflammatory response would allow fungal persistence in the host granuloma without leading to effective fungal elimination or host tissue damage. Based on the integrated theory of microbial pathogenesis "Damage-Response Framework" (DRF) [91–93], we believe that, by reconciling pathogen persistence with host homeostasis, dormancy induction in *C. neoformans* may explain why latent infection is the most common result of the primary infection [17,73].

The inflammasome is an important component of the immune response to fungal pathogens, including *C. neoformans* [94]. Recently, it has been demonstrated that *C. neoformans* secretes small molecules that inhibit the NLRP3 inflammasome in macrophages [57]. Nevertheless, the polysaccharide capsule, structure maintained in VBNC cells [14], is considered the main fungal inhibitory factor, masking cell wall PRRs involved in inflammasome priming [56,95]. Our results corroborate this high capacity of *C. neoformans* cells to frustrate priming in macrophages, adding new insights into the underlying inhibitory molecular mechanism including repression of *Nlrp3* and *Il1b* genes. Importantly, VBNC cells not only share this mechanism of priming inhibition, but also, contrary to non-dormant fungus, does not trigger the NLRP3 inflammasome in primed macrophages. This may reflect its reduced metabolism and non-proliferative state, as indicated by the comparable results obtained with heat-killed STAT cells, and suggests reduced cell host damage potential. The lack of the two signals required for inflammasome activation gains even more relevance considering the hypoxic environment of host granulomas [96,97], where both signals can be potentiated [98–101].

Similarly, dormant *M. tuberculosis* showed a reduced ability to induce IL-1β secretion in macrophages differentiated from the human monocyte cell line THP-1 and peripheral blood

mononucleated cells (PBMC) [102,103]. In addition, the higher the proportion of VBNC cells in biofilms of the opportunistic bacteria *Staphylococcus epidermidis*, the lower the levels of this cytokine in co-cultures with murine BMDMs [5]. In filamentous fungi such as *Aspergillus fumigatus* and *Fonsecaea pedrosoi*, spores–another dormant fungal phenotype–do not activate the NLRP3 inflammasome in THP-1 cells, unlike hyphae fragments [104,105]. We corroborate that microbial dormancy is associated with a limited capacity to activate the inflammasome. Hence, it may serve as a conserved strategy for evading the immune response in latent infections.

As a niche for fungal persistence [28], it is reasonable to expect that phagocytes are also the niche from which *C. neoformans* emerges from dormancy. Hommel et al. [14] did not detect reactivation of VBNC cells inside J774 cells. However, they observed partial reactivation upon incubation with J774 cell lysate or one of its components, namely pantothenic acid (PA). We tested a range of M1/M2 polarized BMDMs, which led to the same result of J774 cells infection until we focused our analysis on the extracellular yeast subset. Hence, we sought to investigate if VBNC cells are not able to reactivate intracellularly because the phagolysosome limits access to reactivation-promoting stimuli such as PA.

Garfoot et al. [65] reported that the phagolysosome lacks biotin and riboflavin, as the proliferation of *Histoplasma capsulatum* auxotrophic mutants for these vitamins is severely compromised in macrophages. Likewise, the survival of those mutants, as well as that of a pantothenate synthase mutant, is impaired *in vivo*, suggesting that PA could also be scarce [65]. In contrast, we detected a comparable intracellular growth of a PA-auxotrophic *C. neoformans* mutant and its parental strain, suggesting the presence of this vitamin or its derivatives in the *C. neoformans*-containing phagolysosome. This observation is in line with previous findings indicating that *C. neoformans*, like other pathogens [106,107], possesses the ability to manipulate the micronutrient composition of the phagolysosome, as described for iron [108,109] and copper [110]. In bacteria, PA-auxotroph attenuates virulence in *M. tuberculosis* but not in *Salmonella* [111,112], reinforcing existing variations in the composition of the phagolysosome depending on the hosted pathogen. *C. neoformans* infection is known to cause phagolysosome membrane permeabilization [35,113–115], which potentially enables access to PA from the host cell cytosol. Moreover, metabolomic evidence in both human and mouse cells indicates that, especially under specific cellular stress conditions, lysosomes can serve as storage sites for amino acids and metabolites, including pantothenic acid, at concentrations reaching up to the micromolar range [116,117]. For instance, one could expect a consequential impact on phagolysosome composition and, in turn, on the maintenance of the intracellular lifestyle of dormant *C. neoformans*. It is noteworthy that the requirement of PA and other vitamins may differ among species, which could explain the divergent results observed for different pathogens [118].

Alternatively, it is conceivable that *C. neoformans* utilizes a derivative of PA within the host phagolysosome. The established alternatives to PA in CoA biosynthesis involve the utilization of 4'-phosphopantetheine as a nexus metabolite [119,120]. Notably, this specific PA derivative is both membrane-permeable and stable in serum. However, the uptake of this intermediate in CoA biosynthesis primarily depends on the degradation of extracellular CoA by heat-unstable enzymes present in serum [121,122]. In addition to the scarce presence of endogenous CoA and 4'-phosphopantetheine in serum [121,123,124], our experiments, conducted exclusively with heat-inactivated FBS, support the presence of PA, rather than a PA derivative, in the *C. neoformans*-containing phagolysosome. Nonetheless, it remains to be precisely determined whether *C. neoformans* employs PA and/or its derivatives during intracellular residence, considering possible implications in the development of antifungal drugs. Thus, the availability of PA or its derivatives supplied by the host during infection may pose challenges for the

development of antifungal therapies that inhibit its biosynthetic pathway [118]. Redirecting the focus of such therapies towards downstream targets, such as fungal acetyl-CoA synthetase, may prove more effective [48,125].

As earlier demonstrated by Hommel et al. [14], VBNC cells exhibited reactivation at 12.5 nM of PA, the same minimum concentration of PA required by the pantothenate synthase mutant to grow. Furthermore, PA was not detected in the conditioned medium derived from VBNC cells, suggesting that VBNC cells have a reduced ability to synthesize this vitamin. In line with that, pantothenate synthase is under expressed in VBNC cells as described in the RNAseq data [14]. Thus, pantothenic acid could be a host-derived trigger for *C. neoformans* reactivation. Future research is needed to elucidate how the phagolysosome controls reactivation despite providing VBNC cells of PA or its derivatives.

Interestingly, increased exposure to the phagolysosome inhibited cell reactivation and led to a decrease in culturability of the remaining ~1% culturable subpopulation of the VBNC inoculum, which are believed to be another phenotype known as persister cells [15]. Notably, the decline in culturability was not accompanied by a proportional increase in cell death. Hence, this result can be interpreted in light of the dormancy continuum hypothesis: under prolonged stress, persister cells gradually reach a deeper state of dormancy switching to VBNC cells [2, 126]. In this sense, the phagolysosomal environment, which recapitulates conditions that induce dormancy, may have turned *C. neoformans* persisters into VBNC cells [127–129]. Alanio et al. [46], when infecting J774 macrophages with *C. neoformans*, observed a time-dependent induction of a yeast subpopulation that met the dormancy criteria. Thus, several converging evidence points that macrophages are the host cellular site for the induction and persistence of the dormant fungus.

Remarkably, as previously mentioned, the subpopulation of VBNC cells that exit macrophages showed an increased rate of culturability of at least 4 times higher than the control during interaction with non-polarized (M0) or M2, but not M1 cells. Of interest, macrophages treated with dexamethasone, an immunosuppressive drug that promotes reactivation of latent cryptococcal infection in animal models [23], also allowed reactivation of VBNC cells. Our findings suggest the involvement of resting or M2-polarized macrophages in reactivation of latent cryptococcosis. Accordingly, in a mouse model of chronic cryptococcosis induced with *C. neoformans Δgcs1* strain, the reactivation achieved after administration of the drug FTY720 was associated with the polarization of M2 cells and their disorganization in the granuloma [130]. Thus, the development of mouse models of latent cryptococcosis by implementation of hypo-virulent mutant or clinical strains [130–132]; and/or VBNC cells may help to determine the immunological parameters underlying reactivation of latent cryptococcosis [29,130].

Moreover, we found that either fungal passage in macrophages or, to a lesser extent, exposure to their EVs, are involved in reactivation. *C. neoformans* undergo profound changes during adaptation to the intracellular environment of macrophages [46,133], which may predispose VBNC cells to reactivate. In addition, the phagolysosome may provide a gradual and less deleterious re-exposure of these severely stressed cells [47] to a richer environment, such as the experimental media. Rich media offer free radicals naturally or as byproducts of their metabolization, thus exacerbating oxidative stress [6,134,135]. Indeed, supplementation with antioxidants such as catalase and pyruvate aids in cell recovery of VBNC bacteria [134–136]. Alternatively, medium dilution, as a way to decrease nutrient availability, led to better adaptation to the growth of VBNC *C. neoformans* [14].

Furthermore, we uncovered a pro-pathogen role of EVs from macrophages. Despite the bidirectional traffic of EVs in pathogen-host interplay [137], most works have focused on *C. neoformans*-derived EVs, with mixed roles in cryptococcosis [39, 137–139]. Recently, Zhang et al. [41] showed the capacity of EVs isolated from BMDMs infected with *C. neoformans* to

enhance the anti-cryptococcal function of recipient macrophages. Contrastingly, in our model, EVs isolated from BMDMs and, to a lesser degree, from peritoneal macrophages and human THP-1 monocytes, aid in the direct reactivation of VBNC *C. neoformans*, making macrophage EVs a paradoxical player in *C. neoformans* infection. Despite technically challenging, given the complex composition of EVs [41,137], our future goal is to decipher the specific factors involved in dormancy reactivation by such pivotal cell-to-cell mediator.

The role of macrophages and other mammalian host cells in reactivating VBNC cells has been previously demonstrated for bacteria, such as *Vibrio cholarae* and *Legionella* sp. [140,141]. In addition, environmental predators of *C. neoformans* including *Caenorhabditis elegans* [142] and *Acanthamoeba* spp. [143,144] also showed the capacity to reactivate these prokaryotic VBNC cells. Given the evolutionary role of those fungal predators in the adaptation of *C. neoformans* to mammalian hosts, they could also be involved in the biology of its reactivation from dormancy [145,146]. Our study opens the possibility of the involvement of macrophages in the reactivation and/or maintenance of latent infections caused by other fungal species, such as *H. capsulatum* [9].

*C. neoformans* can escape from macrophages by lytic and predominantly non-lytic exocytosis (NLE) [42,43,147]. The latter is triggered by living yeast cells and by the production of virulence determinants such as capsule, phospholipase B and urease [42,44,148]. Indeed, VBNC cells, which lacked urease activity, showed a considerably lower rate of exocytosis than stationary cells, although higher than heat-inactivated yeast. This regulation of NLE by the pathogen may represent a strategy during latent infection, preserving a reservoir of intracellular yeast cells for more environmentally favorable conditions, such as immune response failure, based on the principles of microbial seed banks [149]. For instance, if a portion of exocytosed cells encounters resistance from the host, a remaining intracellular subpopulation would persist and sustain the infection for subsequent attempts, thereby preventing fungal eradication from host tissues. It also allows cells to reactivate from the intracellular compartment of macrophages in an optimal environment after the macrophage/monocytes have moved from sites where few immune cells could get rid of the extracellular cells. In rats, a chronic model of cryptococcosis, the vast majority of yeast cells is found intracellularly, a pattern that changes upon immunosuppression, followed by disease reactivation [23].

Although subverted by the pathogen to exert pro-pathogen roles such as fungal dissemination [150] and reactivation, as demonstrated here, exocytosis can also be considered a host-driven strategy [150]. This is supported by the increasing list of pathogens and host factors described to be involved with NLE [150]. Therefore, although it seems contradictory that protective Th1 and Th17 cytokines, but not Th2 cytokines, are related to increased rates of NLE [34,37], proinflammatory signaling could aid in displacing VBNC cells from its protective intracellular niche without promoting reactivation. In fact, NLE from IFN-γ/LPS-activated M1 cells thwarts reactivation. Notably, this inhibition occurs despite their isolated EVs being as effective as those derived from M0/M2 cells in reactivating VBNC cells, suggesting that direct interaction with M1 cells may hinder reactivation stimulated by EVs. Experiments are underway to elucidate this inhibitory mechanism, focusing on a potential role of nitric oxide, especially considering that VBNC cells are not able to neutralize this molecule. In line with this hypothesis, Voskuil et al. [151] showed that non-microbicidal nitric oxide levels modulate a dormancy regulon and inhibit respiration in *M. tuberculosis* leading to dormancy.

In conclusion, our results provide evidence that macrophages control the maintenance and reactivation of dormant *C. neoformans* infection, bringing insights into its underlying mechanisms. The study of macrophage-dormant *C. neoformans* interaction may help to better understand the pathogenesis of cryptococcal infection.

## Material and methods

### Ethics statement

Animal experiments conducted at the Institut Pasteur were approved by the Ethics Committee of the Institute and the French Ministry of Agriculture (CETEA 2013–0135) and carried out in compliance with the French and European regulations on animal care (2010/63/EU; French law 2013–118, February 6th, 2013). All animal procedures performed in Brazil were approved by the Animal Ethics Committee of the University of Brasília (UnBDoc n° 66729/2016) and followed the Brazilian Council for the Control of Animal Experimentation (CONCEA) guidelines.

### Yeast culture

*Cryptococcus neoformans* strain H99O (var. *grubii*; originally obtained from J. Heitman, Duke University, NC) stored at -80°C was initially spread on Sabouraud Dextrose Agar (SDA) at 30°C. After 2 to 5 days of growth, yeast cells (around $10^7$ cells) were collected using a loop and suspended in a T25cm$^2$ flask with vented cap containing 10 mL of YPD broth (1% yeast extract, 2% peptone and 2% dextrose) at 150 rpm and 30°C for 22 h. Next, 100 µL of this pre-culture was used as an inoculum for a new flask to another 22 h of incubation to obtain station-ary-phase cells (STAT). Logarithmic-phase cells (LOG) were obtained from 16 h-old precultures. Viable But Non-Culturable (VBNC) cells were generated using the HypNOS pro-tocol (Hyponutrition and Hypoxia for Seven days) [152]. Briefly, cultures at stationary-phase (i.e., depleted of nutrients) were placed in an airtight bag containing a hypoxic atmosphere generator (<0.1% oxygen) (GENbag anaero, Biomérieux) and statically incubated for 7 days in the dark at 30°C. Cultures prepared at the same conditions but under normoxia (NORM) were used as a control. Culturability and viability were monitored in every experiment to con-firm the VBNC phenotype. Yeast cells were washed 2–3 times in PBS, counted in a Neubauer chamber and the yeast suspension adjusted as required.

### Yeast viability analysis by flow cytometry

Yeast cell viability was assessed using fluorescent staining methods coupled to flow cytometry, according to Hommel et al. [14]. Briefly, LIVE/DEAD Fixable Violet (LVD) (405 nm/450 nm) or Green (488 nm/530 nm) membrane-impermeant amine-reactive dyes (250 µL at 1:1000; Invitrogen) were incubated with $10^6$–$10^7$ cells for 20 min at 30°C and protected from light. Next, samples were washed in PBS and acquired on the Guava easyCyte 12HT Benchtop sys-tem (Merck). Dead cells, which present loss of membrane integrity, are stronger labeled than live cells, and hence distinguishable from the later. Heat killed cells were used as a control and obtained by incubating STAT cells at 70°C for 1 h (referred to as heat killed or HK-STAT).

### Culturability test

To assess culturability (i.e., capacity of a yeast cell to form a colony), yeast cells recovered from hypoxia or from interaction with macrophages were washed in PBS and counted using the Guava easyCyte 12HT Benchtop flow cytometer (Merck) or a hemocytometer (Neubauer chamber). Subsequently, the yeast suspension was adjusted to $10^4$ and $10^5$ cells/mL PBS, and 100 µL of this suspension was inoculated in duplicate onto SDA plates. Seeded plates were incubated at 30°C for 3–5 days to count colony forming units (CFU). Results were expressed as mean percentage of culturability or, alternatively, as "relative culturability" between experi-mental groups at 24 h time point and the inoculum control at 2 h time point.

## Growth curves

Yeast growth curves were generated and analyzed as described by Hommel et al. [14] and Agrawal et al. [152], with adaptations. Briefly, yeast suspensions in minimal medium (200 μL final volume/well containing $2 \times 10^3$ cells) were incubated with the following components: a serial ten-fold dilution of pantothenic acid (D-Pantothenic acid hemicalcium salt; Sigma), 10 μg/mL (protein content) of macrophage-derived extracellular vesicles, 50% conditioned medium (v/v) from *C. neoformans* strain H99O, 1 mM of lysine, leucine, pyruvic acid, or tween 80, 10 mM sodium acetate, 25 mM sodium citrate, or 0.1% Bovine Serum Albumin (BSA). Microplates were incubated for 5 days at 30°C and high agitation. Optical density (OD) at 600 nm was measured every 20 min using Bioscreen C (Oy Growth Curves) or Eon (BioTek) spectrophotometers. To prevent interference caused by condensation buildup on the microplate lid, the lids were treated with a hydrophobic coating (0.05% Triton X-100 in 20% ethanol) according to Brewster [153]. After normalization of each OD value to the initial OD, growth curves were created and analyzed in Graphpad Prism software v9 for the determination of latency of growth (lag or λ phase).

## Cell wall monosaccharide composition

Yeast cell wall fractionation and analysis were performed as previously described [154] with minor adaptations. Briefly, 2 technical replicate cultures of *C. neoformans* were pooled, washed exhaustively in PBS, and boiled in Tris-EDTA-SDS-β-mercaptoethanol buffer (pH 7.4) for 1 h, twice. After centrifugation, the pellet (cell wall) was water-washed extensively, lyophilized and the weight was recorded. Dried cell wall was then subjected to alkali-fractionation with 1 M NaOH and 0.5 M $NaBH_4$ for 1 h at 70°C, twice. The alkali-insoluble fraction (AI) was neutralized by extensive washings, whilst the alkali-soluble fraction (AS) was dialyzed; both the fractions were freeze-dried. AI and AS fractions thus obtained were then hydrolyzed in 4 N trifluoroacetic acid or 8 N HCl (for amino-sugars) for 4 h at 100°C, reduced with sodium borohydride and acetylated. Acetylated monosaccharides were extracted with chloroform, dissolved in methanol, and subjected to gas-chromatography to determine monosaccharides composition in the cell wall; meso-inositol was used as an internal standard.

## β-glucan exposure

*C. neoformans* and, as a control, *Candida albicans* reference strain SC5314, were cultured as abovementioned for analysis of β-glucan exposure on cell surface. Yeast cells were incubated in blocking buffer (PBS plus 2% FBS) containing a soluble human Fc-conjugated dectin-1 (s-dectin-1) at 5 μg/mL for 90 min at room temperature. After washing, cells were resuspended in buffer with secondary antibody human Fc-specific IgG-FITC (1:100) for 40 min. Cells were washed again and acquired by flow cytometry. As a control, cells were stripped of capsule by treatment with 10 mL DMSO for 15 min with stirring before labeling.

## Urease activity

Yeast cells were cultured as aforementioned and incubated in a 96-well plate at $10^8$ cells/mL of Christiansen's urea broth at 37°C and 100 rpm for 2 h. Cell supernatant was obtained after plate centrifugation, transferred to new wells, and photographed for visual inspection and read at 550 nm for OD measurement in a spectrophotometer. Color transition from yellow to pink and increased OD were indicative of urease activity.

## Detection of pantothenic acid in conditioned medium

STAT and VBNC cells were incubated at a density of $10^6$ or $10^8$ cells/mL of minimal medium (MM) at 30°C and 150 rpm for 4 h. Following incubation, the cultures were centrifuged, and the resulting supernatant was filtered through a 0.22 μm membrane to obtain cell-free conditioned medium (CM). The *C. neoformans* 07414Δ mutant strain (obtained from the Hiten Madhani collection) was then incubated in MM, either supplemented or not, with 50% fresh CM (v/v) derived from STAT or VBNC cells (STAT-CM or VBNC-CM, respectively). The growth of the mutant was monitored by measuring the OD at 600 nm using a spectrophotometer. Rise in OD was indicative of the positive presence of pantothenic acid.

## Generation of bone marrow-derived macrophages (BMDM)

Bone marrow-derived macrophages (BMDMs) were generated by a previously described protocol [155]. Briefly, bone marrow cells were isolated by flushing femurs and tibias of 10–12 weeks old wild-type (WT) C57BL/6 or NLRP3$^{-/-}$ male mice with ice-cold RPMI-1640 medium (Gibco). Debris and erythrocytes were removed by passage through a 40-μm cell strainer and treatment with tris-buffered ammonium chloride, respectively. Bone marrow cells were seeded at $2 \times 10^6$ cells/Petri dish in 10 mL RPMI-1640 supplemented with 20% heat-inactivated fetal bovine serum (FBS), 30% L929 cell conditioned medium (LCCM) obtained using the murine L929 fibroblast cell line (ATCC), and 1% penicillin/streptomycin (Invitrogen) and placed in a humid incubator at 5% $CO_2$ atmosphere and 37°C. On day 3, another 10 mL of fresh complete medium was added. On day 7, adherent BMDMs were harvested using TrypLE Express (Thermofisher) and plated at $10^6$ cells/mL overnight in RPMI-1640 supplemented with 10% FBS and 5% LCCM the day before the experiments and referred to here as resting macrophages or M0 cells [64]. As determined by flow cytometry, 90.5% of those cells were CD11b$^+$F4/80$^+$.

## Interaction assays of BMDM and *C. neoformans*

After overnight adhesion, the resting BMDMs monolayer was washed and incubated at $10^6$ cells/mL of complete experimental medium (RPMI supplemented with 10% FBS, unless stated otherwise) in the presence of 10 μg/mL of opsonizing monoclonal antibody (mAb) 18B7 with or without *C. neoformans* cells at a multiplicity of infection (MOI) of 5. After 2 h of interaction, wells were washed with warm RPMI 2 to 4 times to remove non-internalized fungus and the culture protracted with fresh complete experimental medium up to indicated time points. When specified, macrophages were stimulated with LPS (*Escherichia coli* serotype O111:B4, Sigma) at 500 ng/mL (or less, when indicated), for macrophage activation. LPS was also used alone or concomitantly with Nigericin (20 μM, during the final 40 min of incubation; Invivo-Gen) for inflammasome priming and activation, respectively.

## Nitric oxide (NO) production

NO production was indirectly determined by quantifying nitrite catabolite ($NO_2^-$) by the Griess reaction method. Briefly, samples of cell culture supernatant were incubated with an equal volume of Griess reagent (combination of a 1% sulfanilamide solution in 5% ortho-phosphoric acid and a 0.1% naphthylethylenediamine hydrochloride solution) for 5 min at room temperature. The nitrite concentration was calculated based on a standard curve of sodium nitrite (1.56 to 100 μM) in complete experimental medium. The azo dye formed was quantified by spectrophotometry at 540 nm.

## Cytokine production

Cell supernatants were screened for cytokines using the Bio-Plex Pro Mouse Cytokine 23-plex assay kit coupled to Luminex 200 system, according to the manufacturer's instructions (Bio-Rad). When indicated, levels of IL-1β and IL-6 or pro-IL-1β present in the cell supernatant or cell lysate samples, respectively, were assessed by the enzyme-linked immunosorbent assay (ELISA) according to the manufacturer's guidelines (Invitrogen). Results were expressed in pg/mL of cytokine.

## Reactivation assay of VBNC cells co-cultured with BMDMs

Resting macrophages (M0) were stimulated or not (Uns) with 50 ng/mL of IFN-γ (500 IU; Immunotools) alone or concomitantly with 100 ng/mL of LPS (Sigma-Aldrich), 20 ng/mL of IL-4 (Immunotools) or 0.1 μM dexamethasone (Sigma). After 24 h of treatment, wells were washed and inoculated with yeast cells as aforementioned. As a control, the suspension of fungi and opsonizing antibody (Inoculum) was plated in the absence of phagocytes. After 2 h of incubation, wells were washed, and the co-culture interrupted for analysis (H2 control) or protracted up to 24 h with fresh medium containing stimuli to maintain the macrophage polarization states. Macrophages treated concomitantly with IFN-γ and LPS received a dose of LPS 5 times higher (500 ng/mL) during restimulation.

For recovery of intracellular yeast, macrophages were disrupted with 200 μL of PBS containing 0.05% SDS at room temperature for 1 min. After confirmation of lysis by microscopy, the yeast suspension was added of 800 μL PBS, homogenized and centrifuged at $1\,700 \times g$ for 4 min. Yeast cells were washed twice with PBS and checked for culturability and viability. At the 24 h time point, prior to washing the macrophage monolayer, extracellular yeast cells were collected, washed and also analyzed for culturability but not viability (due to a limited amount of yeast cells recovered). Relative culturability was calculated between the culturability of each experimental group at 24 h time point and the inoculum group at 2 h time point (control). Ratios that crossed a threshold value of 1 were used as an indicator of reactivation. For co-cultures using transwell system (0.4 μm; Thincert, Greiner), the inoculum suspension was adjusted to 0.5 mL RPMI and placed in the upper compartment, while the BMDMs were seeded in 1.5 mL RPMI with 2% FBS on the bottom of a 12-well plate. Transwell inserts were included in all conditions to eliminate potential bias related to transwell system usage, and co-culture was conducted at a lower temperature (30°C) optimal for the fungus, based on Dragotakes et al. [156].

## Real-time PCR

To confirm macrophage polarization, total RNA extraction was performed from stimulated macrophages before the infection for analysis of the expression of signature genes of the M1 (*iNOS*, *Il6*) or M2 profiles (*Arg1*) by RT-qPCR. The cell supernatant was collected for nitrite and IL-6 dosage. In parallel, cells were evaluated for the expression of MHC-II (highly expressed in M1 cells) by flow cytometry. Total macrophage RNA was extracted with the RNAeasy kit (Qiagen) according to the manufacturer's instructions. Transcript levels of *iNOS*, *Il6* and *Arg1* were evaluated by quantitative real-time RT-PCR (RT-qPCR) by the SYBR green method using the SuperScript III Platinum SYBR green one-step qRT-PCR kit (Invitrogen). Reactions were carried out in a 20-μL final volume containing 1x supermix (SYBR Green I, MgSO$_4$, Platinum taq DNA polymerase, dNTPs and stabilizers), 0.3 μM of each primer and 4 ng of template. The thermal cycler program (LightCycler 480; Roche Diagnostics) was as follows: 50°C for 15 min, followed by 5 min denaturation at 95°C and 50 cycles at 95°C for 15 s and 60°C for 30 s. Gene expression was normalized to the constitutive gene *GAPDH* and

expressed as "Fold change", calculated by the $2^{-\Delta\Delta Ct}$ method [157]. The primers used are listed in S3 Table.

## High-throughput RNA sequencing of BMDMs

After 6 h of *C. neoformans*-macrophage interaction (MOI 5), the macrophage monolayer was washed followed by RNA isolation using the RNAeasy kit (Qiagen) according to the manufacturer's recommendations. RNA quantification and integrity were assessed using the Agilent 2100 Bioanalyzer. Polyadenylated RNA-enriched libraries were prepared using the TruSeq Stranded mRNA Sample Prep Kit (Illumina) according to the manufacturer's recommendations. Samples were sequenced with the HiSeq 2500 system (Illumina) at the Biomics Platform facility at the Institut Pasteur (Paris, France), generating around 25 to 30 million reads of 100 bp paired-end fragments per library. Reads were processed to remove adapter sequences and low-quality sequences using cutadapt v1.11. Only sequences with a minimum length of 25 nucleotides were considered for further analysis. Alignment to the reference genome (*Mus musculus* GRCm38 from Ensembl v92) was carried out using STAR v2.7.0a with default parameters. Gene quantification was performed using featureCounts v1.6.1 from the Subread package with the following parameters: -t gene -g gene_id -s 2. Count data were analyzed using the R software v3.5.3 and Bioconductor packages including DESeq2 v1.22.2. Genes were considered as differentially expressed when the Benjamini-Hochberg adjusted p-values were <0.05 and |Log2FC|>0.5. Heatmaps depicting the differentially expressed genes were generated using the gplots package or the GraphPad Prism v7.0 software. Gene ontology biological process of differentially expressed genes was annotated using org.Mm.eg.db package. The area-proportional Venn diagram was produced using an online tool developed by Hulsen et al. (available at https://www.biovenn.nl/index.php) [158].

## Isolation of extracellular vesicles (EVs)

EVs were isolated from BMDMs stimulated with IL-4 (M2), both LPS and IFN-γ (M1), or unstimulated (resting) cells, as aforementioned. EVs were also isolated from other phagocytes, including mouse thioglycolate-elicited peritoneal macrophages (PMs), obtained as resting cells, as detailed elsewhere [159], and cells from the human monocyte cell line THP-1, obtained from Rio de Janeiro Cell Bank (BCRJ; n˚ 0234). To isolate EVs, phagocytes were seeded at $1 \times 10^6$ cells/mL in T182cm² flasks with complete experimental medium, restimulated with their respective stimuli (to maintain polarization of BMDMs) and incubated for 24 h. To eliminate the confounding effects of contaminating EVs derived from the FBS used as supplement in the cell culture from the EVs indeed derived from the BMDMs, the FBS used was previously depleted of EVs by ultrafiltration, accordingly to Kornilov et al. [160], with adaptations. Briefly, heat inactivated FBS (Gibco) was filtered through a 0.22 μm pore size membrane and sequentially ultrafiltered through a 100 kDa pore size membrane using an Amicon system (Merck Millipore). The resultant filtrate was confirmed for depletion of EVs by Micro BCA Protein Assay (Thermo Fisher Scientific).

The conditioned medium obtained from the phagocytes was collected, centrifuged at 300 × *g* and filtered through a 0.45 μm pore size filter to remove cell debris. The resulting filtrate was then ultracentrifuged (100 000 × *g* at 4˚C for 1 h*)*, and the resulting pellet was washed in PBS and ultracentrifuged once again. The pellet consisting of EVs was resuspended in PBS, quantified and stored at -20˚C. EVs were indirectly quantified based on protein content as mentioned before [39] and incubated at 10 μg/mL in minimal medium with VBNC cells for growth kinetics analysis using an Eon BioTek spectrophotometer at 600 nm.

## Intracellular proliferation assay

To track intracellular proliferation, *C. neoformans* wild-type strain KN99α (var. *grubii*) and 07414Δ mutant strain cells were stained with calcofluor white (CFW; 10 μg/mL PBS for 10 min at room temperature). Then, yeast cells were incubated in the presence of mAb 18B7 with or without the murine macrophage-like J774.16 cell line (hereafter J774 cells; ATCC) at a MOI of 2.5 in DMEM added of 10% FBS as described previously [70]. Of note, J774 cells were primed overnight with 500 ng/mL LPS, inasmuch this cell line is more permissive to crypto-coccal proliferation when activated [71]. After 2 h, extracellular yeast cells were washed with warm DMEM and the incubation protracted for 24 or 48 h. Intracellular yeast cells were recov-ered and processed, as already described, for flow cytometry. Alternatively, the experiment was conducted by substituting both J774 cells and DMEM with BMDM cells and RPMI medium, respectively.

## Non-lytic exocytosis (NLE) assay

NLE was analyzed according to Bürgel et al. [57]. Briefly, yeast cells were stained with CFW, opsonized with 20 μg/mL of mAb 18B7 for 30 min and incubated with resting BMDMs (MOI 5). Control groups consisting of heat-killed STAT cells (HK; 70˚C, 1 h) alone or added of 1% live STAT cells were included. After 2 h of interaction, wells were washed several times to remove non-phagocytosed yeast and refilled with experimental media. At indicated time-points, including at 2 h (right after the washing step), extracellular yeast cells were collected from the supernatant, whereas intracellular yeast cells were obtained separately after cell lysis using SDS 0.05%. After each collection, samples were washed, added of an equal volume of 1% paraformaldehyde in PBS and stored at 4˚C before acquisition at a BD LSRFortessa cytometer at 1 min/sample for yeast enumeration.

## Statistical analysis

Statistical analysis was conducted using the GraphPad Prism v7.0 software. The one- or two-way ANOVA tests followed by the Dunnett or Tukey post-test were used for comparisons to a control group or between groups, respectively. Student's t-test was used to compare two inde-pendent groups. P-values less than 0.05 were considered statistically significant.

## Supporting information

**S1 Fig. Heatmap of DEGs related to membrane-bound pattern-recognition receptors (PRRs) signaling pathways obtained from macrophage transcriptome.**
(TIF)

**S2 Fig. Heatmap of DEGs related to the nitric-oxide synthase biosynthetic processing GO term (GO:0051770) enriched in macrophages upon VBNC infection.**
(TIF)

**S3 Fig. Validation of the cell treatments used to activate BMDMs.** (**A**) Median fluorescence intensity (MFI; arbitrary units, a.u.) of BMDMs after indicated treatments showing increased MHC-II expression upon IFN-γ stimulation, which is attenuated by concomitant administra-tion of dexamethasone (Dex). (**B**) Detection of increased levels of nitrite in the supernatant of BMDMs stimulated with IFN-γ+LPS, which are damped by the co-addition of dexamethasone. Co-stimulation with IFN-γ and LPS enhanced transcription (**C**) and secretion of IL-6 (**D**), as assessed by real-time PCR and ELISA assay, respectively. Addition of dexamethasone decreased IL-6 release. Analyzes were conducted after 24 h of treatment. Il-6 gene

transcription was normalized against the constitutive gene GAPDH and expressed as fold change. Data from 1 representative experiment out of 2. ***p<0.001, between the indicated groups or compared to unstimulated control group (uns).
(TIF)

**S4 Fig. Pantothenic acid (PA) (or its derivatives) is essential for the growth of PA-auxotrophic *C. neoformans*.** (**A**) *C. neoformans* 07414Δ mutant strain was incubated in minimal medium (MM) supplemented with the CoA precursor PA, or with the following primary/secondary precursors of acetyl-CoA: pyruvate, acetate, citrate, ketogenic amino acids (lysine and leucine), tween 80 (as a source of fatty acids), as well as bovine serum albumin (BSA). Growth was solely observed in the presence of PA. Data are means from 1 representative experiment out of 3, conducted with 3–5 technical replicates each.
(TIF)

**S5 Fig. Pantothenic acid (PA) or its derivatives are found within the phagolysosomes of BMDM cells containing *C. neoformans*.** *C. neoformans* wild-type strain KN99α and 07414Δ mutant cells were labeled with calcofluor white (CFW), opsonized, and incubated with LPS-primed BMDM cells (MOI 2.5) in RPMI + 10% FBS. Yeast cells retrieved from the macrophage upon cell lysis at 2 and 24 h of interaction were analyzed by flow cytometry. The percentage of KN99α and 07414Δ cells exhibiting a medium to low CFW signal (indicative of cells that have undergone proliferation) increased in a comparable manner over time, indicating the presence of PA/PA derivatives in the *C. neoformans*-containing phagolysosome. Data are means ± SD from 1 representative experiment out of 2.
(TIF)

**S6 Fig. *C. neoformans* possesses two functional genes involved in transportation of external pantothenic acid (PA).** (**A**, **B**) The *C. neoformans* 07414Δ mutant strain, lacking PA synthase, and the 00540Δ and 07402Δ mutant strains (all obtained from Madhani collection), which lack PA transporter proteins, were incubated in minimal medium (MM) in the presence or absence of PA (125 μM). Addition of PA enabled the growth of the 07414Δ mutant, whereas it increased the growth of 00540 Δ and CNAG_07402 Δ mutants similarly to the wild-type strain (KN99α). Data are means from 1 representative experiment out of 2, conducted with 3 technical replicates each.
(TIF)

**S7 Fig. Resting BMDMs facilitate reactivation of dormant *C. neoformans* through the secretion of extracellular vesicles (EVs).** Latency of growth of VBNC cells incubated in minimal medium (MM) is decreased in the presence of pantothenic acid (PA; 125 μM) or EVs (10 μg/mL) isolated from cultures of resting BMDMs. Data are means ± SD from biological duplicates where each dot depicts 1 technical replicate. ***p<0.001, compared to control group (MM).
(TIF)

**S1 Table. Gene ontology (GO) enrichment analysis for biological process terms obtained from macrophage transcriptome.**
(XLSX)

**S2 Table. Cytokine levels by Luminex assay.**
(XLSX)

**S3 Table. List of oligonucleotide primers used for real-time PCR analysis.**
(XLSX)

## Acknowledgments

We thank Béatrice Poirier-Beaudouin for the technical support with the Luminex reader and Rachel Legendre for help with the RNAseq analysis.

## Author Contributions

**Conceptualization:** Raffael Júnio Araújo de Castro, Anamélia Lorenzetti Bocca, Alexandre Alanio.

**Data curation:** Vishukumar Aimanianda, Hugo Varet, Anamélia Lorenzetti Bocca, Alexandre Alanio.

**Formal analysis:** Raffael Júnio Araújo de Castro, Clara Luna Marina, Christian Hoffmann, Vishukumar Aimanianda, Hugo Varet, Anamélia Lorenzetti Bocca, Alexandre Alanio.

**Investigation:** Raffael Júnio Araújo de Castro, Clara Luna Marina, Aude Sturny-Leclère, Pedro Henrique Bürgel, Sarah Sze Wah Wong, Vishukumar Aimanianda, Ruchi Agrawal.

**Supervision:** Alexandre Alanio.

**Visualization:** Raffael Júnio Araújo de Castro, Christian Hoffmann, Ruchi Agrawal.

**Writing – original draft:** Raffael Júnio Araújo de Castro.

**Writing – review & editing:** Raffael Júnio Araújo de Castro, Christian Hoffmann, Vishukumar Aimanianda, Hugo Varet, Ruchi Agrawal, Anamélia Lorenzetti Bocca, Alexandre Alanio.

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
