## [Decision Letter · Decision Letter 0]

24 Jul 2023

Dear Dr. Alanio,

Thank you very much for submitting your manuscript "Kicking Sleepers Out of Bed: Macrophages Promote Reactivation of Dormant Cryptococcus neoformans by Extracellular Vesicle Release and Non-lytic Exocytosis" for consideration at PLOS Pathogens. As with all papers reviewed by the journal, your manuscript was reviewed by members of the editorial board and by several independent reviewers. In light of the reviews (below this email), we would like to invite the resubmission of a significantly-revised version that takes into account the reviewers' comments.

Both reviewers appreciated the importance of this work and the quality of the data in most parts. However, some areas of these studies could be strengthened to solidify some of the conclusions.

Whenever conclusion cannot be fully justified with additional data, please soften your conclusions and address the limitations of the current approach.

We are looking forward to seeing the revised manuscript.

We cannot make any decision about publication until we have seen the revised manuscript and your response to the reviewers' comments. Your revised manuscript is also likely to be sent to reviewers for further evaluation.

Sincerely,

Michal A Olszewski, DVM, PhD

Section Editor

PLOS Pathogens

Michal Olszewski

Section Editor

PLOS Pathogens

Kasturi Haldar

Editor-in-Chief

PLOS Pathogens

orcid.org/0000-0001-5065-158X

Michael Malim

Editor-in-Chief

PLOS Pathogens

orcid.org/0000-0002-7699-2064

Both reviewers appreciated the importance of this work and the quality of the data in most parts. However, some areas of these studies could be strengthened to solidify some of the conclusions.

Whenever conclusion cannot be fully justified with additional data, please soften your conclusions and address the limitations of the current approach.

We are looking forward to seeing the revised manuscript.

Reviewer's Responses to Questions

**Part I - Summary**

Reviewer #1: The manuscript takes a very interesting and important look at how macrophages may promote reactivation of dormant Cryptococcus neoformans. Reactivation of dormant C. neoformans infections is certainly a very important topic of clinical relevance. Interestingly, the manuscript also provides some interesting information regarding the impact of introducing yeast at the logarithmic or stationary phases along the growth curve on the macrophage response (not the point of the study but interesting, nonetheless). The studies are, overall, well-conceived and have good controls. The VBNC cell model system for investigating dormancy in C. neoformans is good. The studies show that VBNC cells (and yeast grown to the stationary or logarithmic stage) do not induce much, if any, immunostimulatory responses from macrophages. Also, the data suggests that EVs from resting BMDMs may induce reactivation of VBNC cells. A potential weakness of the study is that the observations were limited to BMDMs. Some data showing similar observations using primary murine monocytes/macrophages and human monocytes/macrophages would have increased the relevancy of the study (although one would not expect for all of the studies to be replicated in another cell type). Also, there is some question as to if the model provided in Figure 9D is fully supported by the results provided in the manuscript (perhaps the model can be revised, more results provided, or this reviewer is mistaken). Overall, this is a nice and thought-provoking study.

Reviewer #2: This manuscript is focused on elucidating the mechanisms that allow for reactivation of dormant or viable, but not culturable (VBNC) C. neoformans cells in the context of macrophage phagocytosis. Using the C. neoformans strain H99O, the authors employ a series of in vitro studies to explore the interplay between VBNC cells and macrophages. Their major findings are as follows: 1) VBNC cells are minimally immunostimulatory in bone marrow derived macrophages (BMDMs) both transcriptionally and via cytokine profiling, 2) VBNC cells are not able to activate the NLRP2 inflammasome, and 3) resting and M2-polarized macrophages can induce reactivation of VBNC cells either through extracellular vesicles of non-lytic exocytosis. The experiments focused on understanding how the macrophage, or host, might benefit from VBNC cell expulsion and how it "reactivates" VBNCs are well conducted with appropriate controls.

**Part II – Major Issues: Key Experiments Required for Acceptance**

Reviewer #1: 1. Confirm robustness and human relevancy of observations using primary murine monocytes/macrophages, and human macrophage-like cell lines or PBMCs; respectively. There is no data regarding the promotion of reactivation primary murine cells or human macrophages nor the impact of EVs derived from these cell sources on reactivation.

2. Question remains if exposure to EVs of M1 macrophages allow for reactivation of C. neoformans or is comparable to exposure to EVs from M0 or M2 polarized BMDMs. The studies in figure 9c were done with EVs from non-polarized BMDMs.

3. There were a couple of conclusions described within the proposed model (Figure 9D) that were perhaps not supported by the data shown (or I could have missed it). One is that macrophages can promote induction and maintenance of the dormant state in Cn within the phagolysosome irrespective of the macrophage polarization phenotype (Do the results provided in Figure 6A show that this is not the case with M1 polarized macrophages?). Secondly, the model states that VBNC cells non-lytic exocytosed from MO and M2 cells, but not M1 cells, are prone to reactivation upon exposure to EVs derived from macrophages (I don't think that this was tested using cells exocytosed from M0, M1, or M2 polarized macrophages - perhaps just inferred).

Reviewer #2: The authors also attempt to elucidate how VNBCs are able to maintain such a low profile in macrophages using cell wall analysis, as well as pantothenic acid limitation studies. While interesting, I would argue that these experiments on the pathogen-side are superficial and require additional work to fully develop this aspect of the manuscript.

The use of the 0747delta mutant strain to detect the presence of pantothenic acid (PA) is questionable. The authors do not provide data or evidence that this mutant strain is NOT utilizing an alternative biosynthetic pathway or nutrient to maintain viability within the phagolysosome. In other words, given that C. neoformans is known to utilize alternative nutrients to maintain viability, it's ability to grow in the phagolysosome (a historically PA-poor environment) is not surprising as the phagolysosome may contain an alternative nutrient source that C. neoformans may be utilizing instead.

**Part III – Minor Issues: Editorial and Data Presentation Modifications**

Reviewer #1: (No Response)

Reviewer #2: 1. Figure 7B, and 7D: please revise the coloring of the PA concentration lines. The different shades of blue are not evident upon reviewing a physical copy of the manuscript.

2. Figure 9D: While useful, the proposed model for VBNC cell interaction with macrophages is not easy to interpret. For instance, it is not clear to me whether the PA in the phagolysosome is intended to inhibit or promote reactivation. In addition, the discussion section suggests that M1 macrophages might be purposefully promoting expulsion of VBNCs to remove dormant cells without triggering reactivation. This M1 vs. M2/M0 nuance is missing in the proposed model. More detail is needed.

3. Line 139-140: I'm not sure what the authors are trying to state in this sentence.

4. Line 204-205: The authors do not specify that the transcriptional data of TNFalpha is not shown in any figures. Please either include this data into a pre-existing figure for reference, or provide language that makes this more clear.

5. Line 236-239: The VBNC cells cannot induce IL-1b in any mouse lineage, but unclear if VBNCs also do not induce IL-1b in LPS-primed BMDMs.

6. Line 275: While previous studies show NO production is effective for killing of Cryptococcus, it is unclear to this reviewer that there is a role for NO production in Cryptococcus containment?

7. Line 400-401: Are the authors suggesting EVs from M1 macrophages enhance reactivation at a similar rate to M2/M0 macrophages?

PLOS authors have the option to publish the peer review history of their article (what does this mean?). If published, this will include your full peer review and any attached files.

Reviewer #1: No

Reviewer #2: No
---

## [Decision Letter · Decision Letter 1]

18 Nov 2023

Dear Dr. Alanio,

We are pleased to inform you that your manuscript 'Kicking Sleepers Out of Bed: Macrophages Promote Reactivation of Dormant Cryptococcus neoformans by Extracellular Vesicle Release and Non-lytic Exocytosis' has been provisionally accepted for publication in PLOS Pathogens.

Best regards,

Michal A Olszewski, DVM, PhD

Section Editor

PLOS Pathogens

Michal Olszewski

Section Editor

PLOS Pathogens

Kasturi Haldar

Editor-in-Chief

PLOS Pathogens

orcid.org/0000-0001-5065-158X

Michael Malim

Editor-in-Chief

PLOS Pathogens

orcid.org/0000-0002-7699-2064

Congratulations on the revised manuscript, which we endorse to be accepted.

Reviewer Comments (if any, and for reference):

Reviewer's Responses to Questions

**Part I - Summary**

Reviewer #1: The authors included new experiments, references, and clarifications that have addressed this reviewers concerns. I believe that the subject matter and findings would definitely contribute to our knowledge in the field.

Reviewer #2: The authors appropriately addressed my concerns and I have no additional criticisms.

**Part II – Major Issues: Key Experiments Required for Acceptance**

Reviewer #1: None

Reviewer #2: n/a

**Part III – Minor Issues: Editorial and Data Presentation Modifications**

Reviewer #1: None

Reviewer #2: n/a

PLOS authors have the option to publish the peer review history of their article (what does this mean?). If published, this will include your full peer review and any attached files.

Reviewer #1: No

Reviewer #2: No

---

## [Editor Report · Acceptance letter]

24 Nov 2023

Dear Dr. Alanio,

We are delighted to inform you that your manuscript, "Kicking Sleepers Out of Bed: Macrophages Promote Reactivation of Dormant *Cryptococcus neoformans* by Extracellular Vesicle Release and Non-lytic Exocytosis," has been formally accepted for publication in PLOS Pathogens.

Best regards,

Kasturi Haldar

Editor-in-Chief

PLOS Pathogens

orcid.org/0000-0001-5065-158X

Michael Malim

Editor-in-Chief

PLOS Pathogens

orcid.org/0000-0002-7699-2064